# GELATO: GRAPH EDIT DISTANCE VIA AUTOREGRESSIVE NEURAL COMBINATORIAL OPTIMIZATION

**Paolo Pellizzoni**,\* **Till Hendrik Schulz**,\* **Karsten Borgwardt**
Max Planck Institute of Biochemistry, Martinsried, Germany
{pellizzoni, tschulz, borgwardt}@biochem.mpg.de

## ABSTRACT

The graph edit distance (GED) is a widely used graph dissimilarity measure that quantifies the minimum cost of the edit operations required to transform one graph into another. Computing it, however, involves solving the associated NP-hard graph matching problem. Indeed, exact solvers already struggle to handle graphs with more than 20 nodes and classical heuristics frequently produce suboptimal solutions. This motivates the development of machine learning methods that exploit recurring patterns in problem instances to produce high-quality approximate solutions. In this work, we introduce GELATO, a graph neural network model that constructs GED solutions incrementally by predicting a pair of nodes to be matched at each step. By conditioning each prediction autoregressively on the previous choices, it is able to capture complex structural dependencies. Empirically, GELATO achieves state-of-the-art results, even when generalizing to graphs larger than the ones seen during training, and runs orders of magnitude faster than competing ML-based methods. Moreover, it remains effective even under limited or noisy supervision, alleviating the demand for costly ground-truth generation.

## 1 INTRODUCTION

Graphs are a common representation for structured data, with applications in domains such as chemistry, biology, social networks, and computer vision (Wu et al., 2021). A fundamental challenge in graph analysis is measuring the similarity between two graphs. Perhaps the most natural definition of such (dis-)similarity is the graph edit distance (GED), which quantifies the minimum cumulative cost of transforming one graph into another through a sequence of edit operations, including node and edge insertions, deletions, and substitutions (Bunke & Riesen, 2009; Blumenthal & Gamper, 2017). In practice, GED computations are commonly formulated as a *graph matching* problem, which provides a finite and algorithmically tractable representation of edit paths.

Despite its conceptual appeal, computing the GED is a notoriously NP-hard problem (Bougleux et al., 2017), with exact solvers, often based either on A* search (Riesen et al., 2007) or integer linear programming (Lerouge et al., 2017), struggling to solve instances with more than 20 nodes. Classical heuristics have been proposed to mitigate this challenge, but often trade their computational efficiency with unsatisfactory solution quality (Carletti et al., 2015; Blumenthal et al., 2020).

This difficulty has sparked growing interest in machine learning (ML) approaches to approximate the GED. While earlier works (Bai et al., 2019; Ranjan et al., 2022) focused on regressing the scalar value for the GED, these approaches do not return the edit path or the matching that would yield the predicted distance value, thus lacking interpretability. Moreover, such predicted values might not be integral upper bounds to the GED, and could therefore not be attained by any matching.

More recent work has instead focused on the *combinatorial* nature of the problem, seeking to find good feasible solutions (i.e., edit paths or matchings) to the problem via ML models, thus providing interpretable matchings, and, more importantly, guaranteeing that the provided distances are upper bounds to the true distance. Some models like GENNA* (Wang et al., 2021) and MATA* (Liu et al., 2023b) use ML to guide the A* search, but show limited scalability, hindering their applicability. Other models like GEDGNN (Piao et al., 2023) and GEDHOT (Cheng et al., 2025) take the approach

---

\*Equal contribution.

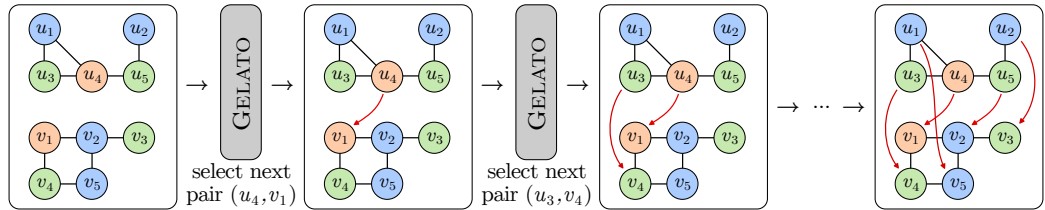

Figure 1: Conceptual visualization of GELATO. Graph matchings are generated in a step-by-step manner. In each step, GELATO is fed autoregressively the previous partial matching, and it predicts the next source-target node pair to be matched, until every source node has been mapped.

of transforming the graph matching problem into some variant of a linear assignment problem, which is polynomially solvable, by learning the assignment costs. This approach is used also in several classical heuristics (Blumenthal et al., 2020). However, since it computes all assignment costs independently from each other, it might miss the interdependencies between node matches.

**Contributions** In this work, we formulate the solution of the GED problem as a sequential decision-making process, where a solution (i.e., a node matching) is constructed incrementally in a step-by-step manner, as visualized in Figure 1. In particular, each intermediate matching defines a subproblem, and our model proceeds in an *autoregressive* fashion: it predicts the next pair of nodes to be matched conditioned on the set of matches predicted so far. This autoregressive process ensures that the decision at each stage leverages both the structural information of the graphs and the history of previous matches, allowing the model to make more accurate predictions. This novel recursive formulation naturally connects GED to the broader field of neural combinatorial optimization (Bengio et al., 2021; Cappart et al., 2023), where learning to exploit structural regularities across problem instances is key to efficiency and generalization (Drakulic et al., 2023).

To this end, we introduce GELATO (Graph Edit distance Learning via Autoregressive neural combinaTorial Optimization), a model that autoregressively predicts node matches using a graph neural network, progressively constructing an approximate solution to the GED instance at hand.

Empirically, we show that GELATO achieves state-of-the-art solution quality on both established and newly introduced benchmarks, and performs well on graphs larger than those seen during training. Moreover, GELATO is substantially faster, often by up to two orders of magnitude, than competing ML-based approaches, and remains effective even when trained with limited or noisy supervision. This robustness directly addresses one of the central bottlenecks in GED research, namely the scarcity of ground-truth matchings, whose exact computation is NP-hard.

## 1.1 ADDITIONAL RELATED WORK

The task of regressing the GED value has been pioneered by SIMGNN (Bai et al., 2019) and later tackled by several models, including GMN (Li et al., 2019), H2MN (Zhang et al., 2021), ERIC (Zhuo & Tan, 2022), GREED (Ranjan et al., 2022), GRAPHEDX (Jain et al., 2024a), and GRAPHSIM (Bai et al., 2020). Beyond the models mentioned in the introduction, several works have tackled the harder yet more informative task of producing GED solutions, such as GOTSIM (Doan et al., 2021) and NOAH (Yang & Zou, 2021). Recently, Verma et al. (2025) introduced GRAIL, an LLM-based model that produces code for GED heuristics.

The *deep graph matching* (Fey et al., 2020) field, albeit related to the GED one, focuses on matching fewer and larger graphs, often coming from computer vision (Gao et al., 2021a; Liu et al., 2025), and usually maximizing the affinity of matched nodes and edges (Yu et al., 2019) rather than minimizing cumulative costs. In this setting, Liu et al. (2023a) introduced a model that predicts node matches sequentially, and is trained with reinforcement learning rather than supervised learning.

In general, *neural combinatorial optimization* focuses on designing machine-learning-based heuristics for NP-hard problems, with examples including TSP (Vinyals et al., 2015; Bello et al., 2016; Khalil et al., 2017), CVRP (Nazari et al., 2018) and the knapsack problem (Drakulic et al., 2023). Approaches span both supervised learning (Fu et al., 2021) and reinforcement learning (Kool et al., 2019; Zhang et al., 2024). Further related work is discussed in Appendix B.

## 2 PRELIMINARIES

A graph is a tuple $G = (V_G, E_G, \ell_G)$, with $V_G$ a finite set of nodes, $E_G \subseteq \{\{u, v\} : u \neq v \in V_G\}$ a set of undirected edges, and $\ell_G : V_G \cup E_G \to \Sigma$ a function assigning nodes and edges to a label of a finite alphabet $\Sigma$. The neighborhood of a node is given by $\mathcal{N}(v) = \{w \in V_G : \{v, w\} \in E_G\}$.

**Graph Edit Distance**   The *graph edit distance* assesses the similarity between two graphs. In particular, it is computed as the minimum cumulative cost of the edit operations required to transform one graph into another, with edit operations being node and edge insertion, deletion, and substitution, each with associated costs. If the costs are a metric, GED is a metric (Justice & Hero, 2006).

In fact, GED can be equivalently defined as a *graph matching* problem or as a quadratic assignment problem, which is more suitable for being solved algorithmically (Bougleux et al., 2017). We define the set of valid matchings $\mathcal{M}(G_1, G_2) = \{\mu \subseteq V_1 \times V_2 \mid \forall (u, v), (u', v') \in \mu : u = u' \iff v = v'\}$, i.e. a set of node pairs such that no source node and no target node is repeated. Given a valid matching $\mu \in \mathcal{M}(G_1, G_2)$, we call $\mu^- = \{(u, \varepsilon) : u \in V_1, \nexists (u, v) \in \mu\}$ the set of source node deletions and $\mu^+ = \{(\varepsilon, v) : v \in V_2, \nexists (u, v) \in \mu\}$ the set of target node deletions. We denote $\bar{\mu} = \mu \cup \mu^- \cup \mu^+$. We call a node pair $(u, v) \in \mu$ a *match*, to distinguish it from a matching.

**Definition 1** (Graph Edit Distance (Bougleux et al., 2017)). *Let $G_1$ and $G_2$ be graphs. The* graph edit distance *(GED) between $G_1$ and $G_2$ is defined by $GED(G_1, G_2) = \min_{\mu \in \mathcal{M}(G_1, G_2)} c(\mu)$, with*

$$c(\mu) = \sum_{(u,v) \in \bar{\mu}} c_n(G_1, G_2, u, v) + \sum_{\substack{(u,v),(w,z) \in \bar{\mu} \\ u < w}} c_e(G_1, G_2, (u, w), (v, z)),$$

*where $c_n$ is the cost function for node edit operations and $c_e$ for edge edit operations (see Def. 3).*

**Graph neural networks**   Message passing graph neural networks (GNNs), for a given graph $G$, produce for each node $v \in V_G$, at each layer $\ell = 1, \ldots, \mathcal{L}$, the embeddings $h_v^\ell \in \mathbb{R}^{d_\ell}$ by taking into account *messages* coming from its neighbors $\mathcal{N}(v)$. More formally, the embedding of node $v$ is updated as $h_v^\ell = f_{\text{upd}}\left(h_v^{\ell-1}, f_{\text{agg}}\left(\{\!\{h_u^{\ell-1} : u \in \mathcal{N}(v)\}\!\}\right)\right)$, where $f_{\text{agg}}$ and $f_{\text{upd}}$ are the aggregate and the update operations, respectively. The first layer of the GNN is fed with the initial node embeddings $h_v^0$, e.g., one-hot encodings of the node labels.

## 3 GRAPH EDIT DISTANCE AS A DECISION PROCESS

In this section, we highlight how the graph edit distance problem can be naturally cast as a *sequential decision process* (Bellman, 1954), and how this perspective can be exploited in the design of machine learning models. This connects the GED problem with a broader line of work on neural combinatorial optimization, where problems are framed as sequential or Markov decision processes in order to leverage policy-based models and dynamic search strategies (Drakulic et al., 2023).

In particular, a sequential decision process in the context of combinatorial optimization is a discrete and deterministic framework for solving problems where a solution is constructed step by step. At each step, the system is in a state $s \in \mathcal{S}$ representing a partial solution, and the algorithm selects an action $a \in \mathcal{A}(s)$ from the feasible set of actions. The action maps the current state to a successor state via a transition function $(s, a) \mapsto s'$. The objective is to construct a sequence of actions that leads from an initial state to a terminal state while minimizing the cost of the terminal state. This formulation mirrors the principles of classical *dynamic programming*, where optimal solutions are obtained by decomposing the problem into subproblems and reasoning recursively over states.

Intuitively, the GED problem solution can be formulated as a sequential decision process by progressively building the matching between the source and target graph. At each step, one selects a node of the source graph and assigns it to a node of the target graph, while ensuring consistency with the partial matching constructed so far. To define this formally, we first introduce a generalization of the GED problem where some matches are already fixed.

**Definition 2** (Graph edit distance with fixed matches). *Let $G_1, G_2$ be graphs. Let $\mu \in \mathcal{M}(G_1, G_2)$ be a matching between $G_1$ and $G_2$. Then the* graph edit distance with fixed matches *(GEDFM) problem on $(G_1, G_2, \mu)$ asks to find $\mu^* = \arg \min_{\nu \in \mathcal{M}(G_1, G_2) : \mu \subseteq \nu} c(\nu)$. We denote the value of the optimal solution as* $\text{GEDFM}(G_1, G_2, \mu) = c(\mu^*)$.

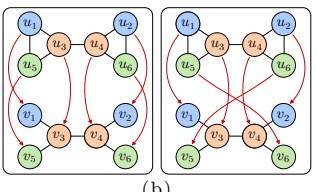 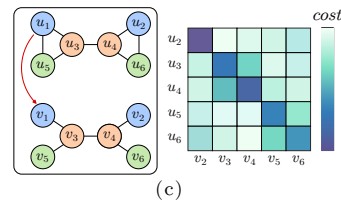

Figure 2: **(a)** A pair of graphs and a possible surrogate cost matrix for node pairs. Entries with the same color must have the same value, as their node pairs are indistinguishable. **(b)** The two node matchings based on the matrix in panel (a) have the same linear-assignment cost, but the one on the right is suboptimal due to inconsistent matches, such as $(u_1, v_1)$ and $(u_5, v_6)$. This happens because the linear assignment cannot capture pairwise dependencies between matches. **(c)** Once the match $(u_1, v_1)$ is fixed and interpreted as an auxiliary edge, pairs $(u_5, v_5)$ and $(u_5, v_6)$ are not indistinguishable anymore. Thus, autoregressive models can make a more informed choice.

To formulate the GED problem as a decision process, we define the state space $\mathcal{S}$ as the set of GEDFM instances. In particular, for the GED problem on input graphs $G_1, G_2$, we define the initial state as the GEDFM instance $(G_1, G_2, \emptyset)$. Note that each state $s = (G_1, G_2, \mu)$, even if $\mu$ leaves some nodes unmatched, represents a valid solution to the matching problem between $G_1$ and $G_2$. Indeed, the unmatched nodes belong to the set of source and target deletions $\mu^+$ and $\mu^-$. Therefore, in our search space, any such state is also a valid terminal state. Then, in a state $s = (G_1, G_2, \mu)$, the set of actions is any pair of unmatched source and target nodes, i.e. $A(G_1, G_2, \mu) = \{(u, v) \in V_1 \times V_2 : \nexists(w, z) \in \mu \text{ with } w = u \text{ or } z = v\}$. This defines the recursion

$$\text{GEDFM}(G_1, G_2, \mu) = \min\left\{c(\mu), \min_{(u,v) \in A(G_1, G_2, \mu)} \text{GEDFM}(G_1, G_2, \mu \cup \{(u, v)\})\right\}.$$

This recursive formulation mirrors the structure exploited in classical search-based approaches such as A*. A straightforward constructive heuristic would be to select, at each state, a single action guided by a surrogate scoring function. In our method, we replace hand-designed surrogates with a learned policy, by predicting the next action directly from the current state using a GNN. This casts GED as a sequential prediction problem, where the node matching is built autoregressively and each decision is conditioned on previous ones, which allows capturing dependencies between matches.

A class of non-sequential methods is that of linear-assignment-based heuristic algorithms, which include, with some slight variations, both classical heuristics such as NODE and BRANCH (Blumenthal et al., 2020), but also several ML-based ones such as GEDGNN (Piao et al., 2023) and GOTSIM (Doan et al., 2021). These algorithms first compute a surrogate cost associated with matching each node $u \in V_1$ to $v \in V_2$, obtaining a surrogate cost matrix $C \in \mathbb{R}^{|V_1| \times |V_2|}$. Once this matrix is obtained, the matching is computed by solving the linear assignment problem on $C$. Since $C$ is not updated as the matching progresses, these methods cannot adapt decisions to previous choices and thus fail to capture interdependencies between assignments. Indeed, for certain graph pairs, such as the ones represented in Figure 2, no function based on node orbits can yield a matrix $C$ that guarantees the optimal GED solution, showing that such models are not *expressive* enough.

If, however, we use a sequential method and the surrogate cost matrix is updated after each action, it can use information from earlier matches and capture these dependencies. In fact, Lemma 1 (discussed in Appendix D) shows that there exists a function such that greedily selecting a pair at each step recovers the optimal solution, although computing this function in general is intractable. However, under a restricted distribution of instances (e.g., small molecules, which have recurring substructures (Pellizzoni et al., 2025)), we can *learn* such a function with a ML model. This insight is the core of our model's architecture.

## 3.1 AN EFFICIENT STATE-SPACE FOR GED WITH PARTIAL MATCHES

Having formalized the GED problem as a sequential decision process, we address the problem of exploring its state space efficiently. The core principle in the dynamic programming paradigm is to exploit overlapping subproblems to reduce the search space. Crucially, each subproblem must be represented in an efficient and concise way to obtain a reduction of the search space.

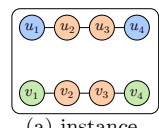
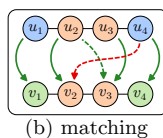

Figure 3: Two different GEDFM instances are mapped via reduce to the same, smaller subproblem, thus reducing the search space.

(a) instance   (b) matching

Figure 4: Solid arrows show an optimal $\mu^*$. While $(u_2, v_3) \notin \mu^*$, it is automorphic to $(u_2, v_2) \in \mu^*$, and hence included in $M^+$. $(u_4, v_2)$ is not automorphic to any pair in $\mu^*$, and is thus in $M^-$.

In our case, for each partial solution for the GED problem (i.e., a GEDFM instance), we construct a possibly smaller GEDFM instance that corresponds to the same subproblem. Since multiple partial solutions will be mapped to the same subproblem, we effectively reduce the size of the search space.

Intuitively, we reduce the graphs in a given instance by removing all nodes that cannot influence the subsequent matches. Clearly, if a node has not been matched yet, it cannot be removed. Similarly, if a matched node $u$ has an unmatched neighbor $v$, then $u$ cannot be removed, as it may influence the choice of the match for $v$. However, if a node and all of its neighbors are matched, it can no longer directly influence any other match, and therefore, it can be removed. In particular, consider a GEDFM instance $(G_1, G_2, \mu)$ and let $\mu^{\text{reduce}} \subseteq \mu$ be all pairs $(u, v)$ for which all neighbors of $u$ or all neighbors of $v$ are matched. Formally, $\mu^{\text{reduce}} = \big\{ (u, v) \in \mu : \forall w \in \mathcal{N}(u), \exists (w, z) \in \mu \text{ or } \forall z \in \mathcal{N}(v), \exists (w, z) \in \mu \big\}$. Remove from $G_1$ all nodes $u$ with $(u, v) \in \mu^{\text{reduce}}$ (and their incident edges), and analogously from $G_2$ all nodes $v$ with $(u, v) \in \mu^{\text{reduce}}$. Let the resulting graphs be $G_1'$ and $G_2'$, and let $\mu' = \mu \setminus \mu^{\text{reduce}}$. Then, we denote the constructed instance $(G_1', G_2', \mu') = \text{reduce}(G_1, G_2, \mu)$.

The following result shows that the optimal solutions for the original and reduced instances are equivalent, meaning that if we restrict an optimal matching $\mu^*$ of an instance to the nodes that are not deleted in the reduction, it remains an optimal matching for the reduced instance.

**Theorem 1.** *Let $(G_1, G_2, \mu)$ be a GEDFM instance and $\mu^* \supseteq \mu$ an optimal matching for it. Then, the instance $(G_1', G_2', \mu') = \text{reduce}(G_1, G_2, \mu)$ has an optimal matching $\mu^* \cap (V_1' \times V_2')$.*

Figure 3 depicts two distinct GEDFM instances that are equivalent to the same subproblem, which is a smaller GEDFM instance. Here, the nodes $u_1$, $u_2$, $v_1$, $v_2$ are deleted from the graphs in the instance on the left, which yields the reduced instance at the center. Similarly, the deletion of $w_1$, $w_2$, $w_3$, $z_1$, $z_2$ and $z_3$ from the instance on the right yields (possibly after renaming the nodes) the reduced instance in the center, even though the original instances were different. In fact, this reduction procedure generally leads to a considerably smaller state-space compared to naively representing GEDFM instances. Feeding the reduced instances to our ML model then enables it to learn from a more compact instance space, possibly making the learning process easier, as formalized in (Drakulic et al., 2023, Section 3).

## 4    GELATO: AN AUTOREGRESSIVE MODEL FOR GED PREDICTION

In this section, we describe the architecture of our model, GELATO (Graph Edit distance Learning via Autoregressive neural combinaTorial Optimization). In a nutshell, the model is a GNN that takes as input a graph representation for a GEDFM instance, and outputs a prediction for an action to be taken, i.e. a pair of yet-unmatched nodes to be matched. This is trained as a link prediction task, with ground-truth given by the optimal graph matchings on the training instances. Then, at inference time, a sequence of actions is predicted autoregressively until the matching is completed. This matching is the output of the model. The next sections detail each of the components.

### 4.1    FROM GED SUBPROBLEMS TO GRAPHS

In order to process GEDFM instances with a GNN model, we encode them as graphs. In particular, let $(G_1, G_2, \mu)$ be a GEDFM instance. Let $H = (V_H, E_H, \ell_H)$ be a graph obtained as the disjoint union of the two graphs, with nodes of $G_1$ labeled with an additional label src, nodes of $G_2$ labeled

with a label $\mathrm{trg}$. Moreover, it contains additional edges $(u, v) \in V_1 \times V_2$ for each match $(u, v) \in \mu$, which take part in the message passing. Practical implementation details are reported in Section F.2.

In fact, we can show (Lemma 2) that the map $(G_1, G_2, \mu) \mapsto H$ is injective. Therefore, a graph-encoded instance discards no information, and is sufficient to make accurate predictions. In practice, we encode the $\mathrm{src}$ and $\mathrm{trg}$ labels as 2-dimensional one-hot encodings appended to node features. Moreover, we label edge-encoded matches with a label $\mathrm{match}$, encoded as a binary flag appended to the edge attribute vectors, to facilitate the decoding process. Finally, given a GEDFM instance, we remark that we first transform it into its reduced subproblem, as described in Section 3.1, and then into its graph-encoded representation before feeding it to the action-predictor model.

## 4.2 MODEL ARCHITECTURE

The action predictor is implemented as a GNN model for link prediction tasks. First, node and edge labels are embedded via a linear layer to vectors $h_u^0$ and $e_{u,v}$ in a $d$-dimensional space. We then use GINE message passing layers (Hu et al., 2020a), as they can take into account also edge features, and they are provably as expressive as 1-WL (Xu et al., 2018). Moreover, we endow each layer with batch normalization (Ioffe & Szegedy, 2015) and residual connections, as they have been shown to be beneficial to the predictive performance of GNN models (Luo et al., 2025). In particular, we have

$$h_u^{\ell+1} = h_u^\ell + \mathrm{ReLU}\Big(\mathrm{BatchNorm}\Big(\mathrm{MLP}\Big(h_u^\ell + \sum_{v \in \mathcal{N}(u)} \mathrm{ReLU}(h_v^\ell + e_{v,u})\Big)\Big)\Big) \in \mathbb{R}^d.$$

Once the node-level representations are obtained after $\mathcal{L}$ layers, one can obtain a scalar representation for a pair of (source and target) nodes by concatenating them and feeding them to an MLP as follows: $o_{u,v} = \mathrm{MLP}\big(h_u^{\mathcal{L}} \| h_v^{\mathcal{L}}\big)$. This can be interpreted as the logit of the likelihood of the match $(u, v)$ to be chosen as the next action in the decision process.

## 4.3 TRAINING

The goal of the model is, given a (graph-encoded) GEDFM instance $(G_1, G_2, \mu)$, to predict the next pair of nodes to match. To do so, the model is trained as a link prediction task (Zhang & Chen, 2018), where each link is formed by a pair of unmatched source and target nodes.

In particular, the model is trained, for a training instance $(G_1, G_2, \mu)$ with ground-truth matching $\mu^*$, to give a high likelihood to any pair of nodes in the optimal matching that is not yet included in the current matching, i.e., $\mu^* \setminus \mu$. We thus don't make the model learn a pre-specified order in which to perform actions: any matching that belongs to $\mu^* \setminus \mu$ is regarded as a valid action.

**Ground truth construction**  Given a set of graph pairs, we obtain the optimal matchings using an exact solver such as MIP-F2 (Lerouge et al., 2017). Then, for each such pair $(G_1, G_2)$, we construct multiple training instances by sampling a set $U$ of source graph nodes, and letting the partial matching be $\mu = \{(u, v) \in \mu^* : u \in U\}$. Since the optimal solution for the instance $(G_1, G_2, \mu)$ is still $\mu^*$, this allows us to generate several training instances for each graph pair while computing the optimal solution only once, which automatically yields a data-augmenting effect.

**Dealing with automorphic matches**  An issue that might arise during training is that conflicts may occur if two node pairs are indistinguishable, but only one is part of the optimal matching $\mu^*$. Since indistinguishable pairs must be assigned identical embeddings, this conflict provides contradictory supervision to the model. For instance, consider the graphs of Figure 4 where the optimal matching $\mu^*$ is depicted by solid green arrows. While the node pair $(u_2, v_3)$ is not in $\mu^*$ and would thus naively be considered a negative edge, it is indistinguishable from $(u_2, v_2) \in \mu^*$. Formally, this indistinguishability is caused by automorphisms. See Appendix C for a formal definition. However, we can show (Lemma 3) that if a pair of nodes $(u, v) \notin \mu^*$ is automorphic to a pair $(w, z) \in \mu^*$, denoted here by $(u, v) \sim (w, z)$, then there exists another optimal solution $\nu^*$ such that $(u, v) \in \nu^*$. Therefore $(u, v)$ should be considered a positive link. Then, we take positive link set as $M^+ := (\mu^* \setminus \mu)/\sim$, i.e., each pair-automorphism class in the ground truth is added once, and the negative set as $M^- := (A(G_1, G_2, \mu)/\sim) \setminus M^+$, i.e. each pair-automorphism class that is not in the positive set is added once.

## 4.4 INFERENCE

At inference, we pursue a sequential process as described in Section 3. Given two graphs $G_1$ and $G_2$, the process starts with GED instance $(G_1, G_2, \emptyset)$. At each step, likelihoods are computed via the GNN model for all unmatched node pairs $(u, v)$, and the pair with the highest score is selected. Formally, at each step, given a graph-encoded instance $(G_1, G_2, \mu)$, we choose the next match to be $\arg\max_{(u,v) \in A(G_1, G_2, \mu)} o_{u,v}$. Then, we create a new instance $(G_1, G_2, \mu \cup \{(u, v)\})$, which is reduced via the map reduce. This instance is then encoded as a graph, and finally used as the input for the next step. Once all nodes of the smallest graph are matched, the process is completed.

While a greedy strategy as the one described above can give satisfactory results, it is very sensitive to the choice of the first match. Indeed, while in later steps the model is informed by previous matches (e.g., on average, neighbors of matched nodes are more likely to be matched), in the first step there is no matching information to exploit. Therefore, to explore the solution space more broadly, we use a simple ensembling strategy that considers multiple starting pairs. More precisely, at the initial instance $(G_1, G_2, \emptyset)$, we select the top-$k$ scoring node pairs (with respect to $o_{u,v}$), and use each such seed to initialize a separate search branch. In fact, to avoid selecting redundant pairs, we select the top-$k$ up to automorphisms. Clearly, $k = 1$ reduces to the standard greedy strategy.

## 5 EXPERIMENTAL EVALUATION

In this section, we experimentally evaluate the predictive performance of GELATO on a set of novel benchmark datasets, which we provide publicly and in an easy-to-use format alongside our code at `https://github.com/BorgwardtLab/Gelato`.

**Datasets** We consider the established SimGNN (Bai et al., 2019) datasets AIDS, LINUX and IMDB, restricted to graphs up to 16 nodes. Moreover, we generate three additional datasets, taking all graphs from the ZINC, molhiv and code2 datasets (Gómez-Bombarelli et al., 2018; Hu et al., 2020b), with up to 16, 16 and 22 nodes respectively. This includes datasets with no labels, only node labels, and both edge and node labels. We randomly split the graphs into training, validation, and test sets using a 60:20:20 ratio. Following Jain et al. (2024b); Roy et al. (2025), who identified that the established GED benchmark datasets are polluted by isomorphic graphs, we ensure that no two graphs across different splits are isomorphic, to prevent data leakage. Moreover, for ZINC and molhiv we additionally create a second test set "`larger`" with graphs from 17 to 24 nodes, and for code2 from 23 to 30 nodes. For each split, we generate graph pairs by randomly sampling from the available graphs, and compute the ground-truth GEDs and corresponding matchings using MIP-F2 (Lerouge et al., 2017). All experiments use uniform edit costs, and we report additional experiments with different edit costs in Appendix H. Further details on datasets can be found in Appendix E.

We emphasize that our datasets, constructed to avoid data leakage and providing ground-truth solutions for larger graphs, offer a *fairer evaluation benchmark* for both this work and future research.

**Evaluation metrics** We evaluate performance using two metrics: normalized mean absolute error (nMAE) and exact hit rate (EHR). The nMAE measures the average error relative to the true values, defined as $^1/_N \sum_{i=1}^N \frac{|\hat{y}_i - y_i|}{\max(1, y_i)}$ where $y_i$ denotes the true GED value and $\hat{y}_i$ the predicted value. For methods returning matchings, we have that $\hat{y}_i - y_i \geq 0$, and the metric measures the optimality gap. The EHR defines the fraction of predictions that exactly match the ground-truth GED value.

**Baselines** We evaluate a diverse set of baseline methods, including classical and learning-based approaches. Among the classical methods, we consider BRANCH, REFINE (Blumenthal et al., 2020), and MIP-F2 (Lerouge et al., 2017) with a timelimit of 0.1s. For learning-based approaches, we consider the best-performing ones, including GEDGNN (Piao et al., 2023), GEDHOT (Cheng et al., 2025), and MATA* (Liu et al., 2023b), which produce (hard) matchings, as well as GREED (Ranjan et al., 2022) and GRAPHEDX (Jain et al., 2024a), which do not. We re-train all learning-based approaches on our datasets using the default settings. Finally, we consider the recent GRAIL (Verma et al., 2025) approach, which leverages large language models to generate code for GED heuristics.

For GELATO, we fix the number of GNN layers to $\mathcal{L} = 5$ and the embedding dimension to $d = 128$, which yields a good trade-off between representational power and memory requirements. See Section G for results with different hyperparameters. We train the link prediction task using the cross-entropy loss with the Adam optimizer ($\text{lr} = 10^{-3}$). We use up to $10^5$ randomly selected pairs

| Method | AIDS | | LINUX | | IMDB-16 | | ZINC-16 | | molhiv-16 | | code2-22 | |
|---|---|---|---|---|---|---|---|---|---|---|---|---|
| | nMAE | EHR | nMAE | EHR | nMAE | EHR | nMAE | EHR | nMAE | EHR | nMAE | EHR |
| Mip-F2$_{(0.1s)}$ | $13.0_{\pm0.9}$ | $63.2_{\pm1.4}$ | $0.3_{\pm0.1}$ | $99.5_{\pm0.1}$ | $25.3_{\pm4.8}$ | $73.9_{\pm1.3}$ | $43.7_{\pm0.7}$ | $6.1_{\pm0.7}$ | $24.1_{\pm0.5}$ | $17.3_{\pm0.6}$ | $20.1_{\pm1.0}$ | $41.2_{\pm2.4}$ |
| Branch | $99.0_{\pm2.6}$ | $1.9_{\pm0.3}$ | $62.4_{\pm3.3}$ | $51.1_{\pm1.8}$ | $11.1_{\pm2.3}$ | $84.1_{\pm1.2}$ | $101.5_{\pm1.1}$ | $1.0_{\pm0.3}$ | $62.9_{\pm0.8}$ | $0.4_{\pm0.1}$ | $79.9_{\pm1.5}$ | $7.7_{\pm0.9}$ |
| Refine | $38.3_{\pm1.8}$ | $20.0_{\pm1.0}$ | $56.2_{\pm2.9}$ | $56.6_{\pm1.3}$ | $5.0_{\pm2.1}$ | $93.4_{\pm0.5}$ | $52.6_{\pm3.5}$ | $1.4_{\pm0.2}$ | $26.5_{\pm1.3}$ | $6.2_{\pm0.7}$ | $182.1_{\pm13.6}$ | $5.6_{\pm0.5}$ |
| Greed | $10.3_{\pm0.2}$ | $38.4_{\pm2.3}$ | $11.4_{\pm0.3}$ | $55.6_{\pm2.0}$ | $4.7_{\pm0.3}$ | $64.6_{\pm1.6}$ | $9.7_{\pm0.2}$ | $24.7_{\pm1.4}$ | $9.9_{\pm0.4}$ | $23.9_{\pm1.3}$ | $8.4_{\pm0.3}$ | $36.7_{\pm1.6}$ |
| GraphEdX | $8.0_{\pm0.3}$ | $51.5_{\pm0.6}$ | $13.8_{\pm0.6}$ | $81.4_{\pm0.4}$ | $17.7_{\pm1.8}$ | $49.2_{\pm1.8}$ | $7.7_{\pm0.2}$ | $35.3_{\pm1.1}$ | $9.9_{\pm0.6}$ | $21.9_{\pm0.6}$ | $6.0_{\pm0.8}$ | $69.3_{\pm1.3}$ |
| Mata* | $6.5_{\pm0.7}$ | $66.1_{\pm2.6}$ | $3.7_{\pm0.4}$ | $89.8_{\pm0.9}$ | $0.4_{\pm0.1}$ | $98.2_{\pm0.4}$ | $18.1_{\pm0.6}$ | $10.7_{\pm0.9}$ | $14.1_{\pm0.1}$ | $15.5_{\pm0.6}$ | $5.1_{\pm0.2}$ | $61.1_{\pm0.9}$ |
| GedGNN | $21.1_{\pm0.8}$ | $38.7_{\pm0.9}$ | $6.7_{\pm0.6}$ | $85.8_{\pm0.9}$ | $2.3_{\pm0.5}$ | $95.1_{\pm0.7}$ | $46.6_{\pm1.1}$ | $3.5_{\pm0.6}$ | $32.1_{\pm0.2}$ | $4.8_{\pm0.7}$ | $17.0_{\pm1.0}$ | $44.2_{\pm1.2}$ |
| GedHot | $6.6_{\pm0.3}$ | $66.8_{\pm1.7}$ | $0.6_{\pm0.1}$ | $98.2_{\pm0.4}$ | $0.2_{\pm0.1}$ | $98.8_{\pm0.4}$ | $27.5_{\pm0.6}$ | $7.1_{\pm0.3}$ | $22.5_{\pm0.4}$ | $9.9_{\pm0.4}$ | $9.0_{\pm0.4}$ | $54.6_{\pm0.9}$ |
| Grail | $2.7_{\pm0.1}$ | $82.1_{\pm0.5}$ | $0.0_{\pm0.0}$ | $100.0_{\pm0.0}$ | $0.0_{\pm0.0}$ | $99.9_{\pm0.1}$ | $12.7_{\pm0.4}$ | $21.3_{\pm0.9}$ | $8.5_{\pm0.2}$ | $33.1_{\pm0.9}$ | $5.8_{\pm0.3}$ | $60.4_{\pm0.8}$ |
| Gelato | $\mathbf{0.1_{\pm0.0}}$ | $\mathbf{99.3_{\pm0.3}}$ | $0.1_{\pm0.1}$ | $99.9_{\pm0.1}$ | $0.1_{\pm0.3}$ | $99.9_{\pm0.1}$ | $\mathbf{0.7_{\pm0.1}}$ | $\mathbf{91.1_{\pm1.1}}$ | $\mathbf{0.5_{\pm0.4}}$ | $\mathbf{95.3_{\pm0.8}}$ | $\mathbf{0.6_{\pm0.4}}$ | $\mathbf{95.7_{\pm0.8}}$ |

Table 1: Overall solution quality of methods in terms of nMAE ($\downarrow$) and EHR ($\uparrow$) in %.

of training graphs, subsampling from each 40 random sub-instances, as described in Section 4.3. Model selection is based on validation nMAE. We use ensembling with $k = 32$, unless noted. We report mean and standard deviation over five runs, each using 1000 random pairs of test graphs.

## 5.1 MAIN RESULTS

We start by evaluating the overall solution quality of our approach. Table 1 demonstrates that GELATO consistently outperforms all baseline methods on both considered evaluation metrics, or matches the strongest baselines on datasets where results are close to optimal. The improvements are particularly substantial on AIDS, ZINC-16, molhiv-16, and code2-22, where GELATO achieves nMAE that is one order of magnitude lower than the second-best performing baseline. On LINUX and IMDB-16, the performance is already quite saturated, which limits the scope for further differentiation. Notably, even relatively simple baselines, such as BRANCH and REFINE, are quite competitive on these two benchmarks. This can likely be attributed to the high number of isomorphic graphs in the test sets, as well as a high degree of automorphisms, which increases the likelihood of many matchings being optimal. Since ZINC-16 and molhiv-16 have edge labels, and most ML-based baselines do not explicitly use edge labels, we report additional results on datasets variants with no edge labels in Table 2. The results further confirm GELATO's superior performance.

Finally, we report in Table 3 inference times, in milliseconds per pair. We observe that the runtimes of the heuristics generated by GRAIL vary depending on the code selected for each dataset. Concerning the learning-based methods, MATA* scales exponentially with graph size, while GEDGNN and GEDHOT suffer from high computational demands due to their reliance on linear assignments and ensembling, often exceeding one second per pair. Compared to these learning-based methods, GELATO, due to its GPU-friendly implementation, achieves runtimes two orders of magnitude lower. For a more comprehensive comparison highlighting GELATO's efficiency, we provide the inference runtimes on CPUs in Appendix H, demonstrating that GPU runtimes are up to 10 times faster than those on CPU. Additionally, we report the training runtimes of all learning-based methods, showing that GELATO 's training is faster compared to other methods that generate matchings.

## 5.2 GENERALIZATION TO LARGER GRAPH SIZES

We now assess the generalizing capabilities of GELATO to larger graphs compared to the ones seen during training. This setting reflects the most critical use case for GED prediction models. For small graphs, exact or near-optimal solutions can be obtained using classical algorithms such as MIP-F2, while it becomes computationally infeasible as graph size grows. Thus, the practical value of ML-based approaches lies in their ability to transfer knowledge from small training instances to larger instances, where exact solvers are no longer applicable.

Graphs from the "larger" test sets were grouped based on their node counts, with 500 pairs randomly sampled from each group. We report mean and standard deviations over five runs. To put results into perspective, we also include groups from the (in-distribution) test sets. The results, illustrated in Figure 5, show that while the performance of GELATO decreases with increasing graph sizes, the decline is gradual and does not accelerate compared to the in-distribution sizes. This suggests that the reduction in performance is likely due to the complexity of the problem (e.g. the number of feasible solutions) scaling with the graph size, rather than the model's ability to

| Method | ZINC-16 | | molhiv-16 | |
|---|---|---|---|---|
| | nMAE | EHR | nMAE | EHR |
| MIP-F2$_{(0.1s)}$ | $51.6_{\pm1.0}$ | $4.3_{\pm0.6}$ | $43.2_{\pm1.2}$ | $10.7_{\pm0.7}$ |
| BRANCH | $127.3_{\pm1.3}$ | $1.0_{\pm0.3}$ | $114.0_{\pm1.7}$ | $0.4_{\pm0.2}$ |
| REFINE | $63.3_{\pm4.2}$ | $1.0_{\pm0.3}$ | $45.2_{\pm2.2}$ | $3.1_{\pm0.7}$ |
| GREED | $9.3_{\pm0.3}$ | $29.1_{\pm2.7}$ | $7.2_{\pm0.4}$ | $37.2_{\pm2.3}$ |
| GRAPHEDX | $6.5_{\pm0.2}$ | $45.3_{\pm1.0}$ | $5.7_{\pm0.3}$ | $46.3_{\pm2.5}$ |
| MATA* | $19.3_{\pm0.5}$ | $13.6_{\pm1.1}$ | $19.5_{\pm0.4}$ | $16.0_{\pm0.8}$ |
| GEDGNN | $48.5_{\pm0.9}$ | $4.1_{\pm0.4}$ | $44.0_{\pm1.0}$ | $6.7_{\pm0.7}$ |
| GEDHOT | $22.0_{\pm0.4}$ | $12.9_{\pm0.7}$ | $18.5_{\pm0.7}$ | $21.8_{\pm1.4}$ |
| GRAIL | $8.3_{\pm0.3}$ | $39.6_{\pm1.7}$ | $6.5_{\pm0.2}$ | $52.5_{\pm1.4}$ |
| GELATO | $\mathbf{0.7}_{\pm0.1}$ | $\mathbf{92.3}_{\pm0.7}$ | $\mathbf{0.5}_{\pm0.2}$ | $\mathbf{95.6}_{\pm0.5}$ |

Table 2: Solution quality on edge-unlabeled graphs. Eval. metrics in %.

| Method | AIDS | LINUX | IMDB-16 | ZINC-16 | molhiv-16 | code2-22 |
|---|---|---|---|---|---|---|
| MIP-F2$_{(0.1s)}$ | $100.0_{\pm0.0}$ | $100.0_{\pm0.0}$ | $100.0_{\pm0.0}$ | $100.0_{\pm0.0}$ | $100.0_{\pm0.0}$ | $100.0_{\pm0.0}$ |
| BRANCH | $0.1_{\pm0.0}$ | $0.0_{\pm0.0}$ | $0.1_{\pm0.0}$ | $0.1_{\pm0.0}$ | $0.1_{\pm0.0}$ | $0.3_{\pm0.0}$ |
| REFINE | $0.2_{\pm0.0}$ | $0.1_{\pm0.0}$ | $0.2_{\pm0.0}$ | $0.9_{\pm0.0}$ | $0.8_{\pm0.0}$ | $2.9_{\pm0.0}$ |
| GREED | $0.1_{\pm0.1}$ | $0.1_{\pm0.1}$ | $0.1_{\pm0.1}$ | $0.1_{\pm0.1}$ | $0.1_{\pm0.1}$ | $0.1_{\pm0.1}$ |
| GRAPHEDX | $0.3_{\pm0.2}$ | $0.3_{\pm0.2}$ | $0.3_{\pm0.2}$ | $0.3_{\pm0.2}$ | $0.3_{\pm0.2}$ | $0.3_{\pm0.2}$ |
| MATA* | $6.8_{\pm0.2}$ | $7.9_{\pm0.4}$ | $174.0_{\pm134.3}$ | $24.2_{\pm4.3}$ | $37.5_{\pm9.1}$ | $6329.8_{\pm2724.3}$ |
| GEDGNN | $742.0_{\pm20.3}$ | $350.7_{\pm2.3}$ | $215.4_{\pm9.0}$ | $1383.2_{\pm9.9}$ | $1252.7_{\pm6.8}$ | $2813.9_{\pm30.2}$ |
| GEDHOT | $1210.7_{\pm23.4}$ | $980.5_{\pm29.5}$ | $410.5_{\pm16.5}$ | $2064.7_{\pm15.5}$ | $1954.1_{\pm75.3}$ | $3951.6_{\pm134.6}$ |
| GRAIL | $8.4_{\pm0.1}$ | $8.1_{\pm0.1}$ | $23.4_{\pm0.4}$ | $44.1_{\pm0.2}$ | $18.9_{\pm0.1}$ | $99.6_{\pm4.3}$ |
| GELATO | $3.2_{\pm0.2}$ | $2.9_{\pm0.1}$ | $3.6_{\pm0.2}$ | $4.2_{\pm0.2}$ | $4.1_{\pm0.1}$ | $5.3_{\pm0.2}$ |

Table 3: Average inference runtime per graph pair (ms).

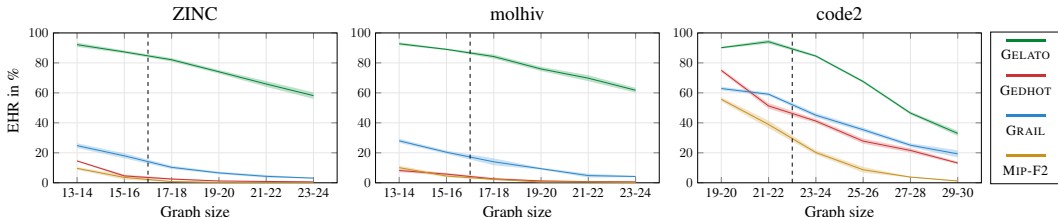

Figure 5: Test solution quality across different graph sizes (EHR in %). Training was conducted on graph pairs up to size 16, 16, and 22. The dashed line separates in- from out-of-distribution data.

generalize. Only results for code2 show a slightly different behavior, as the performance on unseen graph sizes seems to decline at a faster pace, which might be due to a distribution shift in the graphs.

Additionally, the results demonstrate that GELATO consistently outperforms all baselines across all graph sizes, showing its strong performance in both in-distribution and out-of-distribution cases.

## 5.3 ABLATION STUDIES AND ROBUSTNESS TO LIMITED SUPERVISION

In the following, we isolate and assess the impact of key components and parameters of GELATO. Results are reported in Table 4. As a baseline for the ablation studies, we set $k = 1$, reducing inference to the greedy strategy. As expected, this results in a noticeable performance drop compared to GELATO with $k = 32$, highlighting the importance of a broad search space coverage. However, even with $k = 1$, our approach still outperforms all baseline methods in several cases.

First, we ablate the role of *sequential decision making* by predicting the entire matching in one step, instead of generating it autoregressively. Specifically, given two graphs $G_1$ and $G_2$, and pairwise match log-likelihoods between vertices $u \in V_1$ and $v \in V_2$, computed from $o_{u,v}$, we compute the matching corresponding to the optimal linear assignment between $V_1$ and $V_2$. The results (row "No sequential process") show a significant performance drop when this strategy is applied, clearly demonstrating that the autoregressive approach is critical to the performance of GELATO.

Second, we ablate the *instance reduction* discussed in Section 3.1, which shrinks the search space. In this ablation, we omit the reduction step and operate directly on the original, unreduced instances. The results (row "No instance reduction") show that the reduction has an overall positive impact on predictive performance, albeit to a lesser degree compared to the sequential decisions.

Third, we investigate the strategy for selecting positive and negative links in the presence of *automorphic matches*. Recall from Section 4.3, that we choose as positive links the set $(\mu^* \setminus \mu)/\sim$ and ensure that no pair automorphic to any element in $\mu^*$ is contained in the negative link set to avoid contradictions in the training data. Instead, we now consider the naive strategy which for an instance $(G_1, G_2, \mu)$ selects the set $\mu^* \setminus \mu$ as positive links and all others $A(G_1, G_2, \mu) \setminus \mu^*$ as negatives. The results (see row "Automorphic matches") vary across different datasets, with performance deteriorating in some cases, but improving in others. One possible explanation for the observed improvements is that the naive strategy effectively down-weights all automorphic pairs and therefore postpones matching them, which may be beneficial on some datasets. In fact, using the two training strategies as a dataset-specific hyperparameter can yield even better solution quality.

| Method | AIDS | | LINUX | | IMDB-16 | | ZINC-16 | | molhiv-16 | | code2-22 | |
|---|---|---|---|---|---|---|---|---|---|---|---|---|
| | nMAE | EHR | nMAE | EHR | nMAE | EHR | nMAE | EHR | nMAE | EHR | nMAE | EHR |
| GELATO ($k=1$) | $6.4_{\pm0.3}$ | $74.2_{\pm1.0}$ | $9.0_{\pm1.5}$ | $92.8_{\pm0.8}$ | $4.0_{\pm2.2}$ | $98.1_{\pm0.4}$ | $8.6_{\pm0.4}$ | $41.4_{\pm0.6}$ | $5.4_{\pm0.5}$ | $52.8_{\pm1.4}$ | $20.6_{\pm4.5}$ | $65.3_{\pm2.5}$ |
| No sequential process | $61.0_{\pm1.5}$ | $11.6_{\pm0.6}$ | $81.5_{\pm4.3}$ | $60.6_{\pm1.3}$ | $15.9_{\pm3.4}$ | $78.8_{\pm0.9}$ | $70.4_{\pm1.6}$ | $2.0_{\pm0.3}$ | $48.0_{\pm1.0}$ | $2.5_{\pm0.6}$ | $38.8_{\pm0.8}$ | $23.0_{\pm0.8}$ |
| No instance reduction | $8.0_{\pm1.0}$ | $68.7_{\pm0.5}$ | $50.9_{\pm3.0}$ | $84.5_{\pm0.9}$ | $4.5_{\pm2.2}$ | $97.0_{\pm0.4}$ | $9.4_{\pm0.7}$ | $38.2_{\pm2.1}$ | $5.5_{\pm0.3}$ | $51.5_{\pm2.0}$ | $14.5_{\pm1.9}$ | $60.9_{\pm2.4}$ |
| Automorphic matches | $7.5_{\pm1.1}$ | $72.1_{\pm1.8}$ | $16.5_{\pm2.6}$ | $90.0_{\pm1.2}$ | $0.1_{\pm0.0}$ | $99.4_{\pm0.1}$ | $8.7_{\pm0.5}$ | $41.2_{\pm1.1}$ | $6.3_{\pm0.5}$ | $50.5_{\pm0.9}$ | $9.5_{\pm1.4}$ | $70.9_{\pm1.8}$ |

Table 4: Ablation studies of key components of GELATO. Solution quality metrics reported in %.

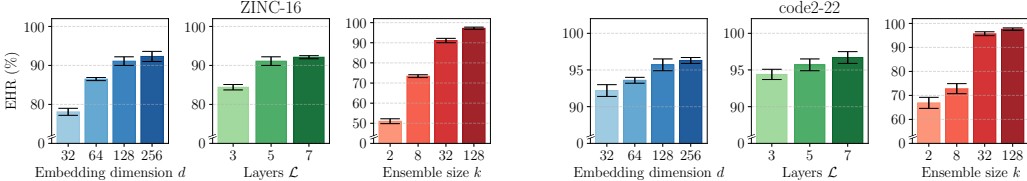

Figure 6: Hyperparameter study on ZINC-16 and code2-22 with varying embedding dimension $d$, number of layers $\mathcal{L}$, and ensembling size $k$, while keeping other parameters at default values $d = 128$, $\mathcal{L} = 5$ and $k = 32$. We report mean and standard deviation over five runs.

Moreover, Figure 6 shows, on the ZINC-16 and code2-22 dataset, how the solution quality varies depending on the embedding dimension $d$, the number of GNN layers $\mathcal{L}$ and the number of ensembles $k$. Results show that lowering the embedding dimension or the number of layers reduces model capacity. On the other hand, increasing them yields modest gains compared to Table 1, at a higher memory cost. Larger ensembles consistently improve results, but increase runtime. Complete results on all datasets are reported in Section G. There, we also show that ablating residual connections and batch normalizations results in a slight decrease in solution quality.

Finally, in Table 5 we investigate the sensitivity of GELATO to limited or noisy supervision. A limiting factor for training GED prediction models is the NP-hardness of computing the ground-truth matchings to be used as supervision signal. Towards this end, we investigate how sensitive GELATO is to being trained on fewer graph pairs, as well as on instances with suboptimal matchings.

On the ZINC and code2 datasets, which are the ones whose ground-truth computation is the most computationally challenging, we train GELATO on only $10^4$ and $10^3$ graph pairs. Moreover, we also train it on $10^4$ graph pairs, where the matchings are computed with MIP-F2 under a

| Data | ZINC-16 | | | code2-22 | | |
|---|---|---|---|---|---|---|
| | Data gen. | nMAE | EHR | Data gen. | nMAE | EHR |
| 10k pairs | 4.1 h | $1.5_{\pm0.2}$ | $83.1_{\pm1.6}$ | 1.9 h | $2.0_{\pm0.9}$ | $90.6_{\pm0.8}$ |
| 10k pairs, $T_{\max} = 1$s | 2.0 h | $1.7_{\pm0.1}$ | $81.6_{\pm0.6}$ | 1.2 h | $1.6_{\pm1.3}$ | $93.3_{\pm1.1}$ |
| 10k pairs, $T_{\max} = 0.1$s | 0.3 h | $3.7_{\pm0.1}$ | $62.6_{\pm0.8}$ | 0.3 h | $1.8_{\pm0.7}$ | $87.3_{\pm0.5}$ |
| 1k pairs | 0.4 h | $5.5_{\pm0.4}$ | $52.1_{\pm2.9}$ | 0.2 h | $4.3_{\pm1.5}$ | $77.5_{\pm0.9}$ |

Table 5: Solution quality with limited supervision

strict timelimit $T_{\max}$ of 1 second and 0.1 seconds, and may thus be suboptimal. While the solution quality degrades, as expected, as the supervision signal worsens, we observe that GELATO still achieves state-of-the-art performance under all weak supervision scenarios, even when training on only $10^3$ pairs, whose generation takes at most 0.4 cpu-hours (i.e., roughly 3 minutes on 8 cores).

# 6 CONCLUSIONS

In this paper we developed GELATO, a state-of-the-art machine learning model for solving the graph edit distance problem, based on a novel autoregressive formulation. However, it could be further improved by designing better inference strategies (e.g., beam search) and hyperparameter search. Moreover, we believe that the field can benefit by focusing on the generalization to larger instances, which can be enabled by the datasets we provide here. Moreover, removing the need for supervision from optimal matchings, which are computationally expensive to compute, would be a significant contribution. This may be achieved using self-improvement (Pirnay & Grimm, 2024) or reinforcement learning, as recently shown for network alignment (Liu et al., 2023a). Finally, machine learning could be leveraged in algorithms with optimality guarantees. For example, integrating GELATO in branch-and-bound methods for exact graph edit distance may reduce the time to reach optimality.

## REPRODUCIBILITY STATEMENT

Code and datasets are available at `https://github.com/BorgwardtLab/Gelato`.

Moreover, the details needed to reproduce our implementation and experimental setup are reported in Appendix F.

## ETHICS STATEMENT

This paper presents work whose goal is to advance the field of machine learning. There are many indirect potential societal consequences of our work, none which we feel must be specifically highlighted here.

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

## A  LIMITATIONS

While our model shows state-of-the-art performance, we believe it can be improved in several ways, which could be addressed by future work. First, GELATO relies on a simple ensembling strategy at inference time, and designing better inference strategies, such as beam search, could lead to significant performance boosts. Moreover, a dataset-specific hyperparameter search for the GNN's architecture and training strategy could further improve results.

Moreover, we believe that a crucial advance in the field would be improving the solution quality on graphs larger than the ones seen at training time. In fact, removing the need for supervision from optimal matchings, which are computationally expensive to compute, would be a possible approach to tackle such a challenge.

## B  ADDITIONAL RELATED WORK

**Graph Edit Distance**  Additional learning-based methods that tackle the GED problem are TAGSIM (Bai & Zhao, 2021), and EGSC (Qin et al., 2021), which address the problem of learning graph dissimilarities, as well as ISONET (Roy et al., 2022), which focuses on graph retrieval. Finally, a recent unpublished preprint proposed DIFFGED (Huang et al., 2025), a diffusion-based model for graph edit distance that shows promising performance. However, their code is not publicly available yet.

Usually edit costs (Definition 3) are considered to be known. In fact, the task of learning the edit costs, pioneered by Neuhaus & Bunke (2004), has been tackled with machine learning in the context of dataset augmentations (Heo et al., 2024), biochemistry (Pellizzoni et al., 2024), and general GED (Leonardi et al., 2025).

**Deep graph matching and network alignment**  The *deep graph matching* (Fey et al., 2020) or *deep network alignment* research field (Emmert-Streib et al., 2016; He et al., 2024) is adjacent to the graph edit distance learning one. Although the graph matching (or network alignment) and the graph edit distance are largely equivalent problems, the focus of learning-based methods to solve the two problems has historically been quite different in terms of objectives, modeling, and scale.

In GED-related works the focus is usually on computing a metric space over several small graphs (e.g., molecules) and on simple edit costs. On the other hand, in works entailing learning-based graph matchings, the focus is usually on fewer and larger graphs, often coming from computer vision (Gao et al., 2021b; Liu et al., 2025), and on more complex costs, usually presented in the form of an affinity matrix (Zanfir & Sminchisescu, 2018; Yu et al., 2019), which possibly depend on continuous node features. Usually, the task is to maximize the affinities of matched node and edges, or to recover some ground-truth alignment between the graphs. Several papers tackled the graph matching problem using deep learning, including approaches based on attention (Yu et al., 2019), on differentiating through combinatorial solvers (Rolínek et al., 2020), on unsupervised methods (Gao et al., 2021a) and on quadratic-assignment-based formulations (Gao et al., 2021b). In this setting, Liu et al. (2023a) introduced the RGM model, which maximizes an affinity score by predicting node matches sequentially, similarly to GELATO. Besides from the different objective function to the one GELATO seeks to optimize, RGM relies on a more computationally expensive product graph, where each node represents pairs of nodes from the original graphs. Moreover, RGM does not seem to exploit any reduction of the search space, as allowed by our Theorem 1. Finally, the model is trained with reinforcement learning rather than supervised learning.

**Neural Combinatorial Optimization**  The neural combinatorial optimization field, pioneered by Vinyals et al. (2015) and Bello et al. (2016), uses deep learning to construct heuristic solutions to NP-hard problems, usually focusing on routing problems such as TSP or CVRP. Approaches include supervised learning methods (Vinyals et al., 2015; Joshi et al., 2019; Fu et al., 2021; Luo et al., 2023), like we do in this paper, but also reinforcement-learning-based methods (Nazari et al., 2018; Kool et al., 2019; Hottung et al., 2022; Zhang et al., 2024; Pirnay & Grimm, 2024). See the review in Bengio et al. (2021) for an overview of the field, and Cappart et al. (2023) for a focus on graph-related problems.

## C ADDITIONAL PRELIMINARIES

In the following, we provide additional definitions that complement Section 2.

### C.1 GRAPH EDIT DISTANCE

Recall that, given two graphs $G_1 = (V_1, E_1, \ell_1)$ and $G_2 = (V_2, E_2, \ell_2)$, the *graph edit distance* (GED) between $G_1$ and $G_2$ is defined by $GED(G_1, G_2) = \min_{\mu \in \mathcal{M}(G_1, G_2)} c(\mu)$, with

$$
c(\mu) = \sum_{(u,v) \in \bar{\mu}} c_n(G_1, G_2, u, v) + \sum_{\substack{(u,v),(w,z) \in \bar{\mu} \\ u < w}} c_e(G_1, G_2, (u, w), (v, z)),
$$

where $c_n$ is the cost function for *node edit* operations and $c_e$ for *edge edit* operations. We then define formally these cost functions.

**Definition 3** (Edit cost functions). *Let $G_1 = (V_1, E_1, \ell_1)$ and $G_2 = (V_2, E_2, \ell_2)$ be graphs. For a pair of nodes $u \in V_1 \cup \{\varepsilon\}$ and $v \in V_2 \cup \{\varepsilon\}$, we define*

$$
c_n(G_1, G_2, u, v) = \begin{cases} 0 & \text{if } u = \varepsilon \text{ and } v = \varepsilon, \\ \delta_{\text{ins}}^{\text{n}}(\ell_2(v)) & \text{if } u = \varepsilon, \ v \in V_2, \\ \delta_{\text{del}}^{\text{n}}(\ell_1(u)) & \text{if } u \in V_1, \ v = \varepsilon, \\ \delta_{\text{sub}}^{\text{n}}(\ell_1(u), \ell_2(v)) & \text{if } u \in V_1, \ v \in V_2, \end{cases}
$$

*where $\delta_{\text{ins}}^{\text{n}}$ is the cost of inserting a node, $\delta_{\text{del}}^{\text{n}}$ the cost of deleting a node, and $\delta_{\text{sub}}^{\text{n}}$ measures the cost of substituting node $u$ with $v$.*

*For a pair of pairs of nodes $(u, w) \in (V_1 \cup \{\varepsilon\})^2$ and $(v, z) \in (V_2 \cup \{\varepsilon\})^2$, we define*

$$
c_e(G_1, G_2, (u, w), (v, z)) = \begin{cases} 0 & \text{if } (u, w) \notin E_1, \ (v, z) \notin E_2, \\ \delta_{\text{ins}}^{\text{e}}(\ell_2(v, z)) & \text{if } (u, w) \notin E_1, \ (v, z) \in E_2, \\ \delta_{\text{del}}^{\text{e}}(\ell_1(u, w)) & \text{if } (u, w) \in E_1, \ (v, z) \notin E_2, \\ \delta_{\text{sub}}^{\text{e}}(\ell_1(u, w), \ell_2(v, z)) & \text{if } (u, w) \in E_1, \ (v, z) \in E_2, \end{cases}
$$

*where $\delta_{\text{ins}}^{\text{e}}$ is the cost of inserting an edge, $\delta_{\text{del}}^{\text{e}}$ the cost of deleting an edge, and $\delta_{\text{sub}}^{\text{e}}$ measures the cost of substituting one edge with another.*

Importantly, in this paper we consider the cost function to be fixed for an entire dataset, i.e., it does not change from one instance to another.

Moreover, we consider only the case in which $\delta_{\text{ins}}^{\text{e}}(\ell_2(v, z)) + \delta_{\text{del}}^{\text{e}}(\ell_1(u, w)) \geq \delta_{\text{sub}}^{\text{e}}(\ell_1(u, w), \ell_2(v, z))$, as this is needed to ensure the equivalence between GED and graph matching (Bougleux et al., 2017, Proposition 1). If this is not the case, one can simply set $\delta_{\text{sub}}^{\text{e}}(\ell_1(u, w), \ell_2(v, z)) = \delta_{\text{ins}}^{\text{e}}(\ell_2(v, z)) + \delta_{\text{del}}^{\text{e}}(\ell_1(u, w))$.

### C.2 ISOMORPHISMS AND AUTOMORPHISMS

We say that two graphs $G$ and $H$ are isomorphic, denoted as $G \simeq H$, if there exists a bijective mapping $\pi : V_G \to V_H$, called isomorphism, such that $\ell_G(v) = \ell_H(\pi(v))$, $\forall v \in V_G$, and $\ell_G((u, v)) = \ell_H((\pi(u), \pi(v)))$, $\forall uv \in E_G$, and $\{\pi(u), \pi(v)\} \in E_H$ if and only if $\{u, v\} \in E_G$.

The group of isomorphisms from $G$ to itself is called the *automorphism* group $\text{Aut}(G)$. Given a graph $G$ and nodes $u, v \in V_G$, we say that they belong to the same orbit if $\exists \pi \in \text{Aut}(G)$ such that $\pi(u) = v$.

Moreover, we can define isomorphisms for graph matchings. In particular, given $G_1, G_2, \mu \in \mathcal{M}(G_1, G_2)$, and $G_1', G_2', \mu' \in \mathcal{M}(G_1', G_2')$, we say that $(G_1, G_2, \mu)$ is isomorphic to $(G_1', G_2', \mu')$ if there exists a bijective mapping $\pi : V_1 \cup V_2 \to V_1' \cup V_2'$ such that (i) $\pi|_{V_1}$ is an isomorphism from $G_1$ to $G_1'$, (ii) $\pi|_{V_2}$ is an isomorphism from $G_2$ to $G_2'$, and (iii) $\{(\pi(u), \pi(v)) : (u, v) \in \mu\} = \mu'$.

Then, we can define the *automorphism* group $\mathrm{Aut}(G_1, G_2, \mu)$ as the group of isomorphisms from $(G_1, G_2, \mu)$ to itself. In particular, given two pairs of nodes $(u, v) \in V_1 \times V_2$ and $(w, z) \in V_1 \times V_2$, we say they are automorphic, denoted as $(G_1, G_2, \mu, u, v) \sim (G_1, G_2, \mu, w, z)$, if $\exists \pi \in \mathrm{Aut}(G_1, G_2, \mu)$ such that $(\pi(u), \pi(v)) = (w, z)$. Given a set of graphs $\mathcal{G}$, we denote the set of pair-orbits as $\mathcal{V}_\mathcal{M} = \{(G_1, G_2, \mu, u, v) \colon G_1, G_2 \in \mathcal{G}, \ \mu \in \mathcal{M}(G_1, G_2), \ u \in V_1, v \in V_2\}/\sim$.

In fact, identifying pair-automorphisms in matchings can be reduced to finding node automorphisms in the graph representation of the instance. For computational efficiency, we use WL classes, enriched with homomorphism counts from small cycles, as a proxy for automorphisms.

## D  PROOFS

**Lemma 1.** *Let $\mathcal{G}$ be a set of graphs. Consider the following algorithm, given a GEDFM instance $(G_1, G_2, \mu)$ and a function $C : \mathcal{V}_\mathcal{M} \to \mathbb{R}$:*

1. *select $(u, v) = \arg\max_{(a,b) \in A(G_1, G_2, \mu)} C(G_1, G_2, \mu, a, b)$;*

2. *set $\mu = \mu \cup \{(u, v)\}$;*

3. *if $A(G_1, G_2, \mu) == \emptyset$ return, otherwise go to step 1*

*Then, there exists a function $C^* : \mathcal{V}_\mathcal{M} \to \mathbb{R}$ such that, for any pair of graphs $G_1, G_2 \in \mathcal{G}$, the algorithm above called with $(G_1, G_2, \emptyset)$ and function $C^*$ outputs an optimal matching $\mu^*$, i.e. such that $c(\mu^*) = GED(G_1, G_2)$.*

*Proof.* We construct such a function. Let $f_1$ be a map $(G_1, G_2, \mu) \mapsto \nu \setminus \mu$ mapping GEDFM instances to the missing matches to get an optimal matching $\nu$. Let then $f_2 = \min \circ f_1$ be a map that selects one match out of the set, based on some arbitrary but universal ordering on matches. Then, we can let $C^*(G_1, G_2, \mu, a, b) = 1$ if $f_2(G_1, G_2, \mu) = (a, b)$, and 0 otherwise.

Then, we have that at each step of the algorithm, a pair $(u, v)$ belonging to an optimal matching is chosen. Therefore, when the algorithm terminates, its output is an optimal matching. $\square$

**Theorem 1.** *Let $(G_1, G_2, \mu)$ be a GEDFM instance and $\mu^* \supseteq \mu$ an optimal matching for it. Then, the instance $(G_1', G_2', \mu') = \mathrm{reduce}(G_1, G_2, \mu)$ has an optimal matching $\mu^* \cap (V_1' \times V_2')$.*

*Proof.* Recall that $\mu^* = \arg\min_{\nu \in \mathcal{M}(G_1, G_2) \colon \mu \subseteq \nu} c(G_1, G_2, \nu)$, with

$$c(G_1, G_2, \nu) = \sum_{(u,v) \in \bar{\nu}} c_n(G_1, G_2, u, v) + \sum_{\substack{(u,v),(w,z) \in \bar{\nu} \\ u < w}} c_e(G_1, G_2, (u, w), (v, z)).$$

We decompose $c(G_1, G_2, \nu)$ by splitting the edge-cost sum into three parts: pairs both in $\bar{\nu} \setminus \mu$, pairs both in $\mu$, and cross-pairs with one element in each. Notice that for the cross-pairs, the first components of $\bar{\nu} \setminus \mu$ and $\mu$ are disjoint in $V_1 \cup \{\varepsilon\}$, i.e. the nodes are either unmatched or matched nodes. Then, the symmetry $c_e(G_1, G_2, (u, w), (v, z)) = c_e(G_1, G_2, (w, u), (z, v))$ lets us combine both orderings, removing the $u < w$ constraint. This gives

$$c(G_1, G_2, \nu) = \sum_{(u,v) \in \bar{\nu}} c_n(G_1, G_2, u, v) + \sum_{\substack{(u,v),(w,z) \in \bar{\nu} \\ u < w}} c_e(G_1, G_2, (u, w), (v, z))$$

$$= \sum_{(u,v) \in \bar{\nu} \setminus \mu} c_n(G_1, G_2, u, v) + \sum_{\substack{(u,v),(w,z) \in \bar{\nu} \setminus \mu \\ u < w}} c_e(G_1, G_2, (u, w), (v, z))$$

$$+ \sum_{(w,z) \in \mu} \sum_{(u,v) \in \bar{\nu} \setminus \mu} c_e(G_1, G_2, (u, w), (v, z)) + C,$$

where

$$C = \sum_{(u,v) \in \mu} c_n(G_1, G_2, u, v) + \sum_{\substack{(u,v),(w,z) \in \mu \\ u < w}} c_e(G_1, G_2, (u, w), (v, z))$$

does not depend on the choice of $\nu$.

Note that for any $(u,v) \in \bar{\nu} \setminus \mu$ both $u$ and $v$ are unmatched, so $u \in V_1'$ and $v \in V_2'$. Therefore, all pairs considered in the first and second sum are in $V_1' \times V_2'$. Note that for $(u,v) \in V_1' \times V_2'$ we have $c_n(G_1, G_2, u, v) = c_n(G_1', G_2', u, v)$, as both nodes are maintained in the reduced graphs. Moreover, for $(u,v), (w,z) \in V_1' \times V_2'$, we have $c_e(G_1, G_2, (u,w), (v,z)) = c_e(G_1', G_2', (u,w), (v,z))$, as both edges are maintained in the reduced graphs.

We now focus on the last sum. In particular, consider a $(w,z) \in \mu \setminus \mu'$. This happens if either $w$ or $z$ have no unmatched neighbors. If all neighbors of $w$ are matched, then for any $(u,v) \in \bar{\nu} \setminus \mu$, since $u$ is unmatched, we have $(u,w) \notin E_1$, so each term $c_e(G_1, G_2, (u,w), (v,z)) = 0$. If instead all neighbors of $z$ are matched, then for any $(u,v) \in \bar{\nu} \setminus \mu$, since $v$ is unmatched, we have $(v,z) \notin E_2$, so $c_e(G_1, G_2, (u,w), (v,z))$ equals $\delta_{\text{del}}^e(\ell_1(u,w))$ if $(u,w) \in E_1$ and 0 otherwise. In both cases, $\sum_{(u,v) \in \bar{\nu} \setminus \mu} c_e(G_1, G_2, (u,w), (v,z))$ is constant in $\nu$.

We can therefore write

$$c(G_1, G_2, \nu) = \sum_{(u,v) \in \bar{\nu} \setminus \mu} c_n(G_1', G_2', u, v) + \sum_{\substack{(u,v),(w,z) \in \bar{\nu} \setminus \mu \\ u < w}} c_e(G_1', G_2', (u,w), (v,z))$$
$$+ \sum_{\substack{(u,v) \in \bar{\nu} \setminus \mu \\ (w,z) \in \mu'}} c_e(G_1', G_2', (u,w), (v,z)) + C'.$$

Given a matching $\nu \in \mathcal{M}(G_1, G_2)$ such that $\mu \subseteq \nu$, let $\nu' = \nu \cap (V_1' \times V_2') \in \mathcal{M}(G_1', G_2')$. Then, we have that $\nu \setminus \mu = \nu' \setminus \mu'$. Since a valid solution to the reduced instance must have $\mu' \subseteq \nu'$, we have that the map $\nu \mapsto \nu'$ is a bijection between valid solutions of the original and the reduced instance.

Consider $\nu' = \nu \cap (V_1' \times V_2') \in \mathcal{M}(G_1', G_2')$. We then have that $\overline{\nu'} = \nu' \cup \{(u, \varepsilon) \colon u \in V_1', \nexists(u,v) \in \nu'\} \cup \{(\varepsilon, v) \colon v \in V_2', \nexists(u,v) \in \nu'\}$. Then, noticing that a node $u \in V_1$ or $v \in V_2$ is unmatched in $\nu$ if and only if $u \in V_1'$ or $v \in V_2'$ respectively is unmatched in $\nu'$, we have that $\bar{\nu} \setminus \mu = \overline{\nu'} \setminus \mu'$.

This lets us write $c(G_1, G_2, \nu)$ only as a function of $\nu'$ as follows:

$$c(G_1, G_2, \nu) = \sum_{(u,v) \in \overline{\nu'} \setminus \mu'} c_n(G_1', G_2', u, v) + \sum_{\substack{(u,v),(w,z) \in \overline{\nu'} \setminus \mu' \\ u < w}} c_e(G_1', G_2', (u,w), (v,z))$$
$$+ \sum_{\substack{(u,v) \in \overline{\nu'} \setminus \mu' \\ (w,z) \in \mu'}} c_e(G_1', G_2', (u,w), (v,z)) + C'.$$

Finally, we turn our attention to $c(G_1', G_2', \nu')$. By the same three-way decomposition,

$$c(G_1', G_2', \nu') = \sum_{(u,v) \in \overline{\nu'}} c_n(G_1', G_2', u, v) + \sum_{\substack{(u,v),(w,z) \in \overline{\nu'} \\ u < w}} c_e(G_1', G_2', (u,w), (v,z))$$
$$= \sum_{(u,v) \in \overline{\nu'} \setminus \mu'} c_n(G_1', G_2', u, v) + \sum_{\substack{(u,v),(w,z) \in \overline{\nu'} \setminus \mu' \\ u < w}} c_e(G_1', G_2', (u,w), (v,z))$$
$$+ \sum_{\substack{(u,v) \in \overline{\nu'} \setminus \mu' \\ (w,z) \in \mu'}} c_e(G_1', G_2', (u,w), (v,z)) + C''.$$

In particular, for any valid instance $\nu \in \mathcal{M}(G_1, G_2)$, we have that $c(G_1, G_2, \nu) = c(G_1', G_2', \nu') + K$, with $K = C' - C''$ a constant not depending on the choice of $\nu$. Then, if $\nu^*$ is an optimal solution to $(G_1, G_2, \mu)$, then $\nu'^* = \nu^* \cap (V_1' \times V_2')$ is an optimal solution to $(G_1', G_2', \mu')$. $\square$

**Lemma 2.** *The map* $\text{encode}(G_1, G_2, \mu) = H$ *is injective and there exists a map* decode *such that* $\text{decode} \circ \text{encode} = \text{id}$.

*Proof.* To prove injectivity it suffices to exhibit the left inverse decode such that decode ∘ encode = id. Given a graph $H = (V_H, E_H, \ell_H)$ that arises from the encoding of $(G_1, G_2, \mu)$, define

$$V_1' := \{ \, v \in V_H \mid \ell_H(v) \text{ marks } v \text{ with src } \},$$

$$V_2' := \{ \, v \in V_H \mid \ell_H(v) \text{ marks } v \text{ with trg } \}.$$

By construction $V_1'$ and $V_2'$ form a partition of $V_H$ into the nodes of $G_1$ and $G_2$ respectively. We then define

$$E_1' := \{ \, e = (u, v) \in E_H \mid u, v \in V_1' \, \},$$

$$E_2' := \{ \, e = (u, v) \in E_H \mid u, v \in V_2' \, \},$$

which by construction correspond to the edges of $G_1$ and $G_2$ respectively. Finally, we recover the matching edges

$$\mu' := \{ \, (u, v) \in E_H \mid u \in V_1', \ v \in V_2' \},$$

which by construction correspond to the node matches in $\mu$. Finally, we let $\ell$ be the function $\ell_H$, without the additional src and trg labels.

Let $G_1' = (V_1', E_1', \ell)$ and $G_2' = (V_2', E_2', \ell)$, and set $\text{decode}(H) = (G_1', G_2', \mu')$.

Therefore encode admits a left inverse and is injective. □

**Lemma 3.** *Let $(G_1, G_2, \mu)$ be a GEDFM instance and $\mu^*$ an optimal solution to it. Let a pair of nodes $(u, v) \in V_1 \times V_2$ such that $(u, v) \notin \mu^*$. Let also $(w, z) \in \mu^*$ be such that $(u, v)$ and $(w, z)$ are automorphic. Then, there exists an optimal solution $\nu^*$ such that $(u, v) \in \nu^*$.*

*Proof.* Since $(u, v)$ and $(w, z)$ are automorphic, there exists a matching automorphism $\pi \in \text{Aut}(G_1, G_2, \mu)$ such that $\pi(w) = u$ and $\pi(z) = v$. Let then $\nu = \{(\pi(a), \pi(b)) \colon (a, b) \in \mu^*\}$. By definition of matching automorphism, we have that $\{(\pi(a), \pi(b)) \colon (a, b) \in \mu\} = \mu$, and since $\mu \subseteq \mu^*$, we have that $\mu \subseteq \nu$. Thus $\nu$ is a feasible solution.

We have $C_n := \sum_{(a,b) \in \nu} c_n(G_1, G_2, a, b) = \sum_{(c,d) \in \mu^*} c_n(G_1, G_2, \pi(c), \pi(d))$. Since $c_n$ is permutation-equivariant, we have $C_n = \sum_{(c,d) \in \mu^*} c_n(\pi^{-1}(G_1), \pi^{-1}(G_2), c, d) = \sum_{(c,d) \in \mu^*} c_n(G_1, G_2, c, d)$, which is the node cost for $\mu^*$. We can obtain the same result for edge costs. Thus, $\mu^*$ and $\nu$ have the same cost, and $\nu$ is optimal.

Finally, since $(w, z) \in \mu^*$, we have that $(u, v) = (\pi(w), \pi(z)) \in \nu$ by construction. □

# E DATASETS

In this section, we discuss details about the datasets we use and introduce in the paper, including instructions on how to download and how to use them.

In our experiments, we consider the established SimGNN (Bai et al., 2019) datasets AIDS, LINUX and IMDB. In particular, AIDS and LINUX contain graphs up to 10 nodes, and thus were provided with ground truths. On the other hand, IMDB contains graphs with up to 50 nodes. In the original datasets, ground truths were provided only for graphs with up to 10 nodes, and suboptimal values for others. Instead, we take the subset of graphs up to 16 nodes, and compute optimal matchings. We denote this dataset as IMDB-16. AIDS is endowed with node labels, but no edge labels. LINUX and IMDB on the other hand have unlabeled node and edges.

Moreover, we generate three additional datasets, taking all graphs from the ZINC, molhiv and code2 datasets (Gómez-Bombarelli et al., 2018; Hu et al., 2020b). Due to computational constraints in computing the optimal matchings, we select the subsets of graphs with up to 16, 16 and 22 nodes, respectively. We denote these datasets as ZINC-16, molhiv-16 and code2-22. code2 is endowed with node labels, but no edge labels. ZINC and molhiv have labels on both node and edges, but we also consider a version with no edge labels.

The graph datasets that we consider cover multiple domains, including molecular graphs (AIDS, molhiv-16, ZINC-16), software networks (LINUX), movie collaboration networks (IMDB-16), and Python program syntax trees (code2-22).

We randomly split the graphs into training, validation, and test sets using a 60:20:20 ratio. Jain et al. (2024a) identified that the established GED benchmark datasets are polluted by isomorphic graphs. Their approach is to remove isomorphic copies of graphs. Instead, we retain all graphs, but we ensure that no two graphs across different splits are isomorphic to prevent data leakage. Retaining isomorphic copies allows us to have a consistent number of graphs with previous works, and to include (both for training and testing purposes) instances where the edit distance is 0.

Table 6 reports some statistics about the datasets. In particular, we report the number of graphs, the number of unique graphs up to isomorphism, the average number of nodes and edges, and the number of node and edge labels. Moreover, we report the number of pairs of graphs for which the optimal GED (with unit costs) and a corresponding matching was computed, as well the average, minimum and maximum value of such GEDs.

| Dataset | # graphs | # iso. classes | # nodes avg. | min. | max. | avg. # edges | # node labels | # edge labels | # pairs | GED avg. | min. | max. |
|---------|----------|----------------|--------------|------|------|--------------|---------------|---------------|---------|----------|------|------|
| AIDS | 700 | 667 | $8.9_{\pm1.4}$ | 2 | 10 | $8.8_{\pm1.8}$ | 29 | - | 245350 | $9.0_{\pm2.7}$ | 0 | 23 |
| LINUX | 1000 | 89 | $7.6_{\pm1.5}$ | 4 | 10 | $6.9_{\pm1.9}$ | - | - | 500500 | $4.7_{\pm2.6}$ | 0 | 16 |
| IMDB-16 | 1185 | 167 | $9.6_{\pm2.5}$ | 7 | 16 | $33.4_{\pm15.9}$ | - | - | 175773 | $22.1_{\pm16.7}$ | 0 | 101 |
| ZINC-16 | 836 | 835 | $14.5_{\pm1.6}$ | 9 | 16 | $15.0_{\pm1.9}$ | 16 | 3 | 349866 | $14.2_{\pm2.7}$ | 0 | 27 |
| molhiv-16 | 6734 | 6733 | $13.5_{\pm2.3}$ | 2 | 16 | $14.1_{\pm2.8}$ | 41 | 4 | 455993 | $16.9_{\pm4.2}$ | 0 | 37 |
| code2-22 | 2087 | 785 | $21.2_{\pm1.2}$ | 11 | 22 | $20.2_{\pm1.2}$ | 58 | - | 435008 | $12.8_{\pm6.9}$ | 0 | 46 |

Table 6: Summary of dataset statistics.

Moreover, for ZINC and molhiv we additionally create a second test set "larger" with graphs from 17 to 24 nodes, and for code2 from 23 to 30 nodes, which are meant to probe the generalization abilities of GED prediction models.

For each dataset and each split, we generate graph pairs by randomly sampling from the available graphs, and compute the ground-truth graph edit distances and corresponding matchings using the MIP-F2 (Lerouge et al., 2017) formulation of GED as an integer linear programming (ILP) problem. The ILPs are solved using the Gurobi (Gurobi Optimization, LLC, 2024) solver, with warm-starts given by the REFINE local search heuristic (Blumenthal et al., 2020).

We make our dataset publicly available and easily installable, in order to offer a fairer evaluation benchmark also for future research. The code can be obtained from GitHub:

```
git clone git@github.com:BorgwardtLab/Gelato.git
```

Then, our dataset class can be easily used, for example, as follows:

```python
from Gelato.src.dataset import GraphMatchingDataset

dataset = GraphMatchingDataset(name='aids', num_pairs=1000, split='test')
for data in dataset:
    print(data.x_s.shape, data.edge_index_s.shape, data.edge_attr_s.shape)  # source graph
    print(data.x_t.shape, data.edge_index_t.shape, data.edge_attr_t.shape)  # target graph
    print(data.matching.shape)                                              # matching
```

## F EXPERIMENTAL SETUP

In this section, we discuss additional details about the experimental setup.

### F.1 BASELINES

For GREED (Ranjan et al., 2022), we used the default parameters suggested by the authors. Specifically, we set the number of GIN layers to 8 with a hidden dimension of $d = 64$. We used all available graph pairs from the training set for training, and, analogously, the complete validation set for validation. GREED minimizes RMSE in the loss function using the Adam optimizer (lr=$10^{-3}$),

and training continued until the pre-implemented early stopping criterion was met. We report the mean and standard deviation over five runs, each using 1000 graph pairs. We utilize the same test sets used as in all other methods. Since GREED outputs continuous regression values, we round the predictions to the nearest integer in case of EHR statistics.

For MATA* (Liu et al., 2023b), we largely adhered to the default parameter settings recommended by the authors. The model consists of three GNN layers (i.e. SEGcn layers), each with a hidden dimension of $d = 64$. The model uses a binary cross-entropy loss function using an Adam optimizer, with a learning rate set to $10^{-3}$. Training was conducted on all available training graph pairs. During training, MATA* employs mini-batch gradient descent with a batch size of 128. Model evaluation was performed every 200 epochs over a total of 10,000 epochs. We chose the model with the best validation performance, measured in terms of nMAE. In all experiments, we set the parameter top-$k = 4$, which governs the number of explored matches for each node. To calculate the GED, we first extracted the matchings considered internally by MATA*. During this process, we modified line 133 in file `Mata.cpp` by swapping the inputs `q` and `g` (i.e., the source and target graphs) in the function call `app->init(q,g,...)`. We believe this was the intended order based on the function's definition and description. Notably, with this adjustment, the results either remained consistent or even showed a significant improvement. Finally, we report the mean and standard deviation over five runs, each using 1000 graph pairs. We use the same test sets considered as in all other methods.

For GEDGNN (Piao et al., 2023), we used the default parameters provided by the original authors. Specifically, the model consists of three GIN layers with hidden dimensions of 64, 32, and 16, respectively. A binary cross-entropy loss is used to minimize the discrepancy between the predicted and ground truth matchings, optimized using Adam optimizer with a learning of $10^{-3}$. We train on all available training graph pairs over a total of 20 epochs. Model selection was based on the best validation performance in terms of nMAE. We set the parameter $k = 100$, which governs the number of considered node matchings and from which the best is finally selected. For evaluation, we report the mean and standard deviation over five runs, each conducted on 1000 graph pairs, using the same test sets as in all other methods.

In case of GEDHOT (Cheng et al., 2025), we adopted the default parameters specified by the authors. Specifically, the model architecture consists of three GIN layers with hidden dimensions of 128, 64, and 32. GEDHOT is an ensemble method that combines GEDIOT and GEDGW. While GEDGW is an unsupervised method that produces matchings via solving a surrogate optimization problem, GEDIOT is trained using a loss function consisting of an MSE loss and a binary cross-entropy loss, which is optimised via the Adam optimizer at a learning rate $10^{-3}$. We train the model on all available graph pairs for 20 epochs and select the best-performing model based on the validation nMAE value. We set the parameter $k = 100$, i.e., the number of considered matchings, from which the best is finally selected. For evaluation, we report the mean and standard deviation over five runs, each using 1000 graph pairs We use the same test sets as for all other methods to ensure a fair comparison.

To evaluate Verma et al. (2025), we used all functions discovered by GRAIL that were provided by the authors. To select the optimal programs, we ran all 43 function ensembles on a validation set of 1000 randomly selected graph pairs and chose the one with the lowest nMAE value for testing. We note that we modified the matching cost computation to correct a rare issue in the original implementation that occurred when singleton nodes were present. Results are reported as the mean and standard deviation over five independent runs, each using 1000 graph pairs, employing the same test sets as for all other methods for consistency.

For GRAPHEDX (Jain et al., 2024a), we followed the parameter choices recommended by the authors. Specifically, we utilized the model designed to take into account node labels. For edge edits, we employed the XOR neural surrogate function, and for node edits, we used AlignDiff, as suggested by the overall best performance in the original study. We trained the model on all available training graph pairs and validated it on all available validation graph pairs. Model selection was based on the best validation performance, measured using nMAE. Training was halted when the validation performance did not improve for 10 consecutive epochs. The model was optimized using Adam with a learning rate of 0.001, minimizing a bounded loss term on edge, node, and label alignment distances. For evaluation, we report the mean and standard deviation over five runs, each conducted on 1000 graph pairs, using the same test sets as those used in all other methods.

### F.2 SETUP FOR GELATO

GELATO is implemented in PyTorch Geometric (Fey & Lenssen, 2019). The action predictor is a GNN model, implemented following the formulas in Section 4.2. At inference time, given an instance $(G_1, G_2, \emptyset)$, we first transform it into a single PyTorch Geometric graph (i.e. as a tuple x, edge_index and edge_attr), following the specification described in Section 4.1. Note that at the first step, no reduction is performed.

Recall that, at each step, given a graph-encoded instance $(G_1, G_2, \mu)$, we choose the next match to be $\arg\max_{(u,v) \in A(G_1, G_2, \mu)} o_{u,v}$ and we create a new instance $(G_1, G_2, \mu \cup \{(u,v)\})$. This is implemented by simply adding the edge $(u, v)$ and its label to the edge_index and edge_attr vectors. Then, the reduction is performed by identifying the nodes that would be deleted by the reduce map, and removing all edges adjacent to them. From the message-passing perspective this is the same as deleting the nodes, but it allows to avoid re-indexing the representation of the graphs at each step. We note that, in the presence of BatchNorm, the embeddings of these isolated nodes can still marginally influence the embeddings of the non-deleted nodes during training (but not during inference), although this does not seem to affect the performance empirically. This reduced graph is then fed to the action-predictor GNN.

In the experiments, unless otherwise specified, we fix the number of GNN layers to $\mathcal{L} = 5$ and the embedding dimension to $d = 128$. We train the link prediction task using the cross-entropy loss with the Adam optimizer, with learning rate $\mathrm{lr} = 10^{-3}$ and no weight decay, for 30 epochs. We use up to $2 \cdot 10^5$ randomly selected pairs of training graphs, subsampling from each 40 random sub-instances, as described in Section 4.3. Model selection is based on the validation nMAE, which is computed at the end of each training epoch. At inference time, we use ensembling with $k = 32$, unless otherwise stated, and the matching attaining the lowest cost is returned.

We now describe the setup for ablation studies. For the "No sequential process" row, we train only with instances with no fixed partial matchings. At inference, we do the following. Given two graphs $G_1$ and $G_2$, we compute the logits $o_{u,v}$ between vertices $u \in V_1$ and $v \in V_2$. We then compute log-likelihoods as $l_{u,v} = \log(\mathrm{sigmoid}(o_{u,v}))$. We then compute the matching corresponding to the optimal linear assignment between $V_1$ and $V_2$, i.e., $\mu_{LA}^* = \arg\max_\mu \sum_{(u,v) \in \mu} l_{u,v}$. Assuming independence between choices, this maximizes the likelihood of the matching. For the "No instance reduction" row, we perform both training and inference without reducing the instances with the map reduce. Finally, for the "Automorphic matches" row, we perform training with the naive strategy which for an instance $(G_1, G_2, \mu)$ selects the set $\mu^* \setminus \mu$ as positive links and all others $A(G_1, G_2, \mu) \setminus \mu^*$ as negatives. Inference is performed in the same way as in the standard model.

### F.3 COMPUTING INFRASTRUCTURE

The experiments are run on a cluster equipped with Intel(R) Xeon(R) Silver 4116 CPUs and NVIDIA H100 GPUs. The code is based on PyTorch and PyTorch-Geometric.

## G HYPERPARAMETER STUDIES FOR GELATO

In this section, we discuss the sensitivity of GELATO to its hyperparameters. In particular, we study the effect of the use of batch normalization and residual connections, as well as the embedding dimension $d$, the number of GNN layers $\mathcal{L}$, and the ensemble size $k$.

First, Figure 7 reports, for all datasets, the EHR ($\uparrow$) when ablating batch normalizations or residual connections from the GNN layers. We fix $d = 128$, $\mathcal{L} = 5$ and $k = 32$, which are the values used for Table 1. Here, we see that the ablation consistently decreases solution quality, albeit with only a minor degradation.

Figure 8 reports, for all datasets, the EHR ($\uparrow$) when varying the embedding dimension $d = \{32, 64, 128, 256\}$, and while keeping $\mathcal{L} = 5$ and $k = 32$, which are the values used for Table 1. The results show that, generally, the solution quality increases with the dimensionality, and that lowering the embedding dimension from the default of $d = 128$ reduces model capacity. On the other hand, increasing the dimensionality to $d = 256$ yields modest gains, but increases the memory

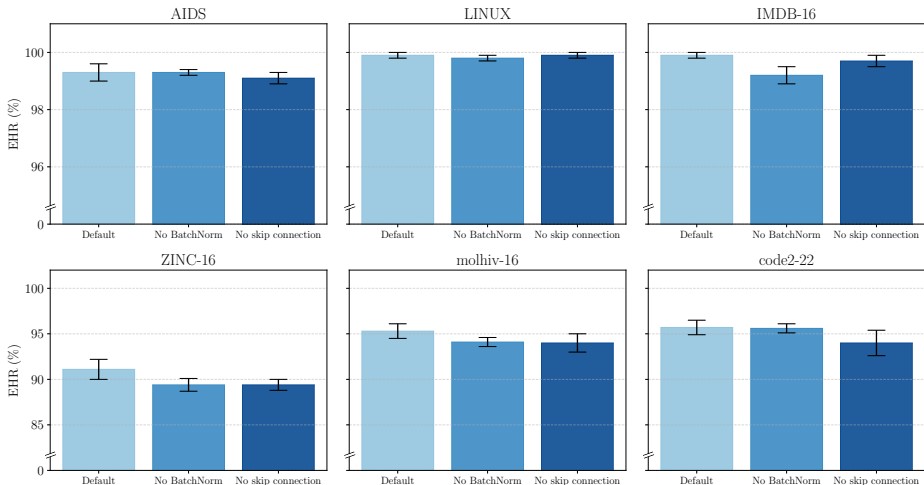

Figure 7: Solution quality for GELATO w.r.t. EHR (↑), ablating the batch normalization and the skip connections in the GNN layers. We report mean and standard deviation over five runs.

requirements (from roughly 200k parameters to 800k) and runtimes (e.g. from $4.2 \pm 0.1$ to $4.8 \pm 0.1$ milliseconds per pair on ZINC).

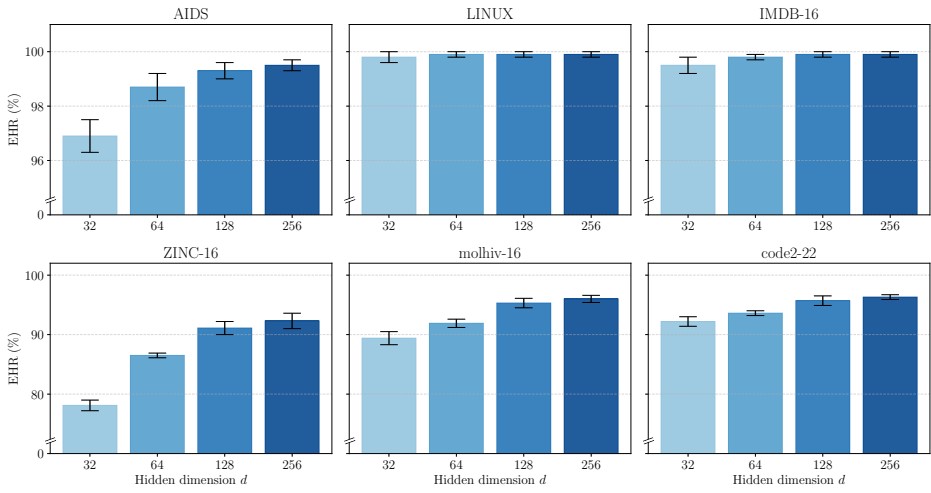

Figure 8: Solution quality for GELATO w.r.t. EHR (↑), varying the embedding dimension $d = \{32, 64, 128, 256\}$ with $\mathcal{L} = 5$ and $k = 32$. We report mean and standard deviation over five runs.

Figure 9 reports, for all datasets, the EHR (↑) when varying the number of GNN layers $\mathcal{L} = \{3, 5, 7\}$, and while keeping $d = 128$ and $k = 32$, which are the values used for Table 1. Similarly to the analysis above, the results show that increasing the number of GNN layers generally improves solution quality, at the expense of increased runtime (e.g. $4.8 \pm 0.1$ milliseconds per pair on ZINC for $\mathcal{L} = 7$) and memory requirements. Once again, moving from $\mathcal{L} = 5$ to $\mathcal{L} = 7$ yields diminishing returns, but decreasing it to $\mathcal{L} = 3$ significantly decreases the capacity. Interestingly, choosing $\mathcal{L} = 1$, i.e. considering only one-hop neighborhoods, drastically reduces the solution quality. For example, on AIDS, the EHR drops to only 13%.

Finally, Figure 10 reports, for all datasets, the EHR (↑) when varying the size of the ensemble $k = \{2, 2^3, 2^5, 2^7\}$, with $d = 128$ and $\mathcal{L} = 5$. As already reported in Table 4, ensembling yields considerable gains to the solution quality, as it alleviates the sensitivity to the choice of the first match. The results show that increasing the number of ensembles monotonically increases the so-

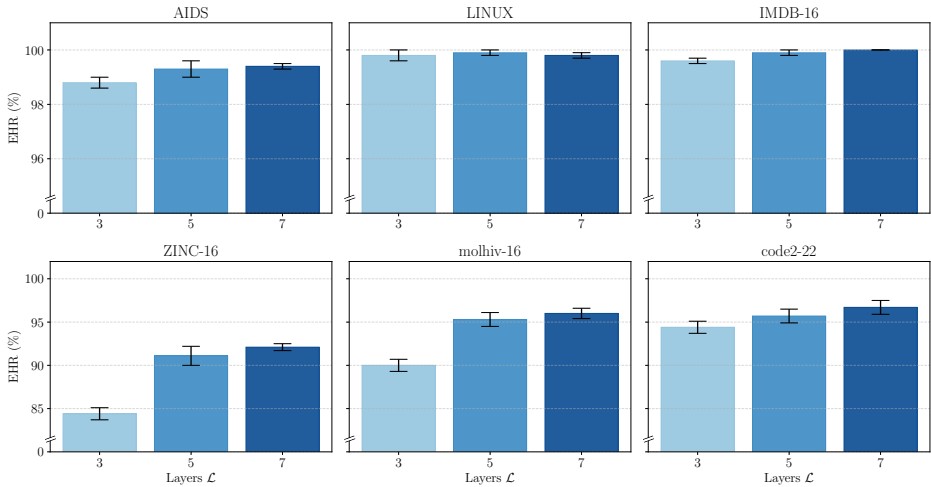

Figure 9: Solution quality for GELATO w.r.t. EHR ($\uparrow$), while varying the number of GNN layers $\mathcal{L} = \{3, 5, 7\}$ with $d = 128$ and $k = 32$. We report mean and standard deviation over five runs.

lution quality, albeit at the cost of increased runtimes. Indeed, increasing $k$ to 128 increases the inference time on ZINC to $6.3 \pm 0.1$ milliseconds per pair. Interestingly, on ZINC, moving from the standard $k = 32$ to $k = 128$ improves the EHR from 91% to 97%, significantly reducing the gap to optimality.

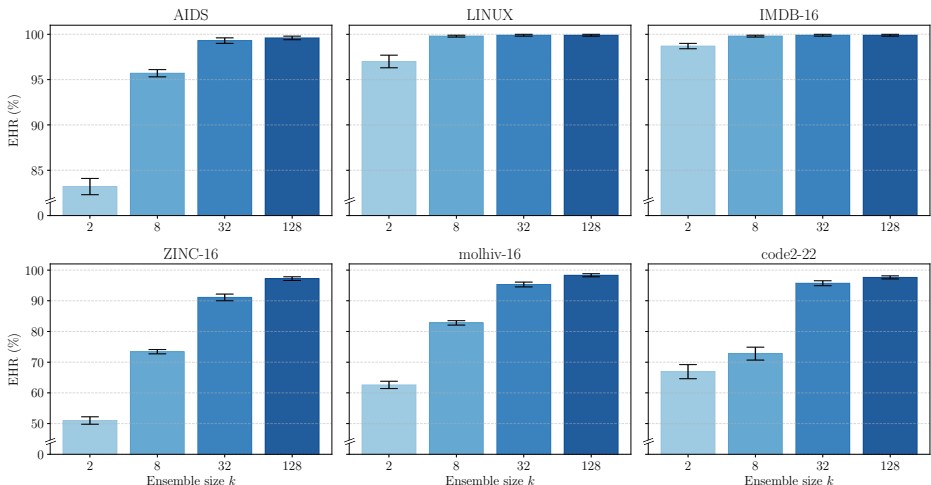

Figure 10: Solution quality for GELATO w.r.t. EHR ($\uparrow$), while varying the size of the ensemble $k = \{2, 2^3, 2^5, 2^7\}$, with $d = 128$, $\mathcal{L} = 5$. We report mean and standard deviation over five runs.

In conclusion, the hyperparameter study shows that the choices of hyperparameters used in the main results are sensible choices that balance good solution quality with memory requirements and runtimes. However, they also showcase that, if better solution quality is required, GELATO can yield even better results by increasing its dimensionality, its number of GNN layers, and the number of ensembles.

## H   ADDITIONAL EXPERIMENTAL RESULTS

In this section, we report additional experimental results that were excluded from the main paper for space constraints.

**Additional metrics**  First, in Table 7 and in Table 8, we report the results of Table 1 using two additional metrics. In particular, we report respectively the mean average error (MAE) and the root mean squared error (RMSE).

The MAE is defined as $1/N \sum_{i=1}^{N} |\hat{y}_i - y_i|$ where $y_i$ denotes the true GED value and $\hat{y}_i$ the value of the predicted matching. For GREED, which does not return a matching, we take $\hat{y}_i$ as the scalar predicted GED. Note that for all methods except GREED, the returned matchings must be upper bounds to the optimal solution, so we have $\hat{y}_i - y_i \geq 0$.

Similarly, the RMSE is defined as $\left(1/N \sum_{i=1}^{N} (\hat{y}_i - y_i)^2\right)^{1/2}$ where $y_i$ denotes the true GED value and $\hat{y}_i$ the value of the predicted matching. For GREED, which does not return a matching, we take $\hat{y}_i$ as the scalar predicted GED.

The results show that, even when evaluating the models using these different metrics, GELATO retains state-of-the-art solution quality, with a wide margin to the second-best performing method, on all datasets except LINUX and IMDB-16. On these two latter datasets, both GRAIL and GELATO provide near-optimal solutions, with the former method yielding slightly better solution quality, as observed in Table 1.

| Method | AIDS | LINUX | IMDB-16 | ZINC-16 | molhiv-16 | code2-22 |
|---|---|---|---|---|---|---|
| GREED | $0.814_{\pm 0.022}$ | $0.485_{\pm 0.014}$ | $1.030_{\pm 0.064}$ | $1.351_{\pm 0.030}$ | $1.537_{\pm 0.034}$ | $1.025_{\pm 0.029}$ |
| GRAPHEDX | $0.586_{\pm 0.013}$ | $0.382_{\pm 0.008}$ | $3.258_{\pm 0.190}$ | $0.92_{\pm 0.015}$ | $1.530_{\pm 0.026}$ | $0.418_{\pm 0.015}$ |
| MATA* | $0.509_{\pm 0.044}$ | $0.244_{\pm 0.026}$ | $0.086_{\pm 0.021}$ | $2.468_{\pm 0.078}$ | $2.170_{\pm 0.034}$ | $0.651_{\pm 0.013}$ |
| GEDGNN | $1.741_{\pm 0.043}$ | $0.394_{\pm 0.031}$ | $0.357_{\pm 0.071}$ | $6.490_{\pm 0.160}$ | $5.035_{\pm 0.031}$ | $2.320_{\pm 0.082}$ |
| GEDHOT | $0.524_{\pm 0.025}$ | $0.037_{\pm 0.007}$ | $0.050_{\pm 0.019}$ | $3.753_{\pm 0.088}$ | $3.365_{\pm 0.037}$ | $0.966_{\pm 0.023}$ |
| GRAIL | $0.224_{\pm 0.008}$ | $0.000_{\pm 0.001}$ | $0.002_{\pm 0.004}$ | $1.759_{\pm 0.055}$ | $1.323_{\pm 0.019}$ | $0.708_{\pm 0.026}$ |
| GELATO | $0.012_{\pm 0.004}$ | $0.002_{\pm 0.002}$ | $0.004_{\pm 0.003}$ | $0.098_{\pm 0.015}$ | $0.053_{\pm 0.009}$ | $0.054_{\pm 0.010}$ |

Table 7: Overall solution quality of methods w.r.t. MAE ($\downarrow$).

| Method | AIDS | LINUX | IMDB-16 | ZINC-16 | molhiv-16 | code2-22 |
|---|---|---|---|---|---|---|
| GREED | $1.029_{\pm 0.017}$ | $0.655_{\pm 0.014}$ | $1.949_{\pm 0.086}$ | $1.719_{\pm 0.027}$ | $2.069_{\pm 0.060}$ | $1.482_{\pm 0.078}$ |
| GRAPHEDX | $0.740_{\pm 0.017}$ | $0.503_{\pm 0.015}$ | $5.942_{\pm 0.267}$ | $1.188_{\pm 0.032}$ | $1.978_{\pm 0.047}$ | $0.579_{\pm 0.025}$ |
| MATA* | $0.966_{\pm 0.048}$ | $0.849_{\pm 0.061}$ | $0.728_{\pm 0.154}$ | $2.941_{\pm 0.078}$ | $2.730_{\pm 0.049}$ | $1.183_{\pm 0.031}$ |
| GEDGNN | $2.697_{\pm 0.070}$ | $1.177_{\pm 0.062}$ | $1.947_{\pm 0.273}$ | $7.371_{\pm 0.131}$ | $5.914_{\pm 0.020}$ | $3.704_{\pm 0.092}$ |
| GEDHOT | $0.998_{\pm 0.025}$ | $0.270_{\pm 0.027}$ | $0.468_{\pm 0.106}$ | $4.441_{\pm 0.109}$ | $4.190_{\pm 0.042}$ | $1.627_{\pm 0.034}$ |
| GRAIL | $0.563_{\pm 0.019}$ | $0.013_{\pm 0.028}$ | $0.061_{\pm 0.086}$ | $2.250_{\pm 0.066}$ | $1.883_{\pm 0.019}$ | $1.293_{\pm 0.044}$ |
| GELATO | $0.167_{\pm 0.047}$ | $0.054_{\pm 0.058}$ | $0.120_{\pm 0.056}$ | $0.339_{\pm 0.034}$ | $0.273_{\pm 0.053}$ | $0.286_{\pm 0.037}$ |

Table 8: Overall solution quality of methods w.r.t. RMSE ($\downarrow$).

Moreover, in Table 9 and Table 10, we report the results on the edge-unlabeled variants of the datasets, as done in Table 2, with respect to MAE and RMSE, respectively. Similarly to the results in the main paper, GELATO achieves the best solution quality also with respect to these two metrics.

Moreover, in Figure 11, we report the results of the experiments on generalization to larger graph sizes, as reported in Figure 5, with respect to nMAE (lower is better). We can see that general trend is the same as reported in Figure 5, with GELATO providing better solution quality compared to all baselines for all instance size ranges.

**Non-unit edit costs**  While in the main paper we performed all experiments with unit edit costs, GELATO can natively support any edit cost function. On the other hand, all other methods that return matchings which were considered as baselines do not seem to support non-unit edit costs.

We report results on AIDS and ZINC-16, using two sets of non-unit edit costs. First, we consider the case where node operations are more expensive than edge operations. In particular, according to

| Method | ZINC-16 | molhiv-16 |
|---|---|---|
| GREED | $1.141_{\pm 0.047}$ | $0.842_{\pm 0.035}$ |
| GRAPHEDX | $0.686_{\pm 0.015}$ | $0.654_{\pm 0.024}$ |
| MATA* | $2.299_{\pm 0.047}$ | $2.302_{\pm 0.025}$ |
| GEDGNN | $5.938_{\pm 0.090}$ | $5.272_{\pm 0.076}$ |
| GEDHOT | $2.614_{\pm 0.046}$ | $2.126_{\pm 0.058}$ |
| GRAIL | $1.011_{\pm 0.034}$ | $0.763_{\pm 0.024}$ |
| GELATO | $0.086_{\pm 0.009}$ | $0.052_{\pm 0.009}$ |

Table 9: Overall solution quality of methods on edge-unlabeled datasets w.r.t. MAE ($\downarrow$).

| Method | ZINC-16 | molhiv-16 |
|---|---|---|
| GREED | $1.447_{\pm 0.044}$ | $1.074_{\pm 0.046}$ |
| GRAPHEDX | $0.875_{\pm 0.022}$ | $0.828_{\pm 0.026}$ |
| MATA* | $2.747_{\pm 0.037}$ | $2.813_{\pm 0.041}$ |
| GEDGNN | $6.799_{\pm 0.094}$ | $6.253_{\pm 0.104}$ |
| GEDHOT | $3.132_{\pm 0.040}$ | $2.731_{\pm 0.064}$ |
| GRAIL | $1.444_{\pm 0.029}$ | $1.221_{\pm 0.033}$ |
| GELATO | $0.340_{\pm 0.042}$ | $0.264_{\pm 0.041}$ |

Table 10: Overall solution quality of methods on edge-unlabeled datasets w.r.t. RMSE ($\downarrow$).

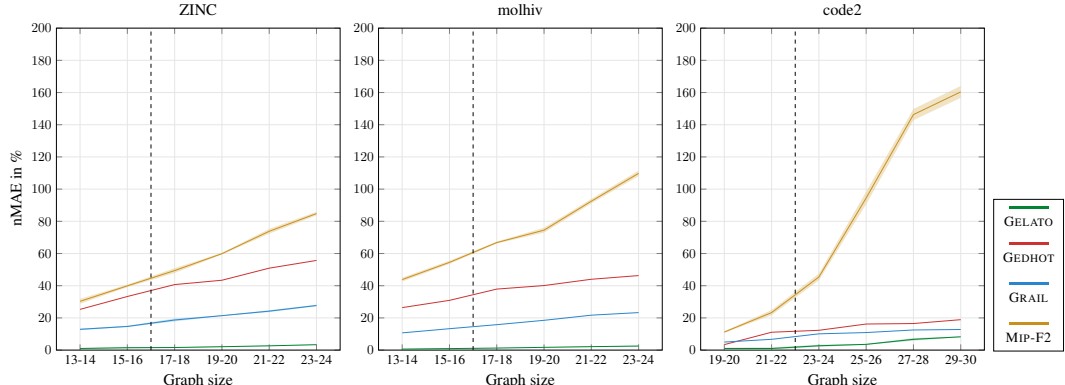

Figure 11: Performance of methods across graph sizes measured by nMAE ($\downarrow$) in %. Training was conducted on graph pairs up to size 16 or 22. The dashed line separates in-distribution from out-of-distribution graphs.

Definition 3, we set $\delta^n_{sub} = \delta^n_{ins} = \delta^n_{del} = 10$ and $\delta^e_{sub} = \delta^e_{ins} = \delta^e_{del} = 1$. For brevity, we denote this $\delta^n = 10$, $\delta^e = 1$. Second, we consider the case where edge operations are more expensive than node operations. In particular, we set $\delta^n = 1$, $\delta^e = 10$.

RESULTS, reported in Table 11, show that GELATO shows state-of-the-art performance also in the presence of non-unit edit costs, with solution quality generally aligning with the solution quality on datasets with unit edit costs.

| Method | AIDS | | | | ZINC-16 | | | |
|---|---|---|---|---|---|---|---|---|
| | $\delta^n = 10,\ \delta^e = 1$ | | $\delta^n = 1,\ \delta^e = 10$ | | $\delta^n = 10,\ \delta^e = 1$ | | $\delta^n = 1,\ \delta^e = 10$ | |
| | nMAE | EHR | nMAE | EHR | nMAE | EHR | nMAE | EHR |
| MIP-F2(0.1s) | $0.8_{\pm 0.1}$ | $89.0_{\pm 1.4}$ | $31.7_{\pm 1.6}$ | $57.6_{\pm 1.9}$ | $8.5_{\pm 0.3}$ | $18.8_{\pm 1.5}$ | $72.5_{\pm 0.9}$ | $3.8_{\pm 0.8}$ |
| BRANCH | $20.0_{\pm 1.2}$ | $3.4_{\pm 0.2}$ | $247.3_{\pm 12.8}$ | $1.8_{\pm 0.3}$ | $24.0_{\pm 0.2}$ | $1.0_{\pm 0.3}$ | $195.9_{\pm 1.0}$ | $1.0_{\pm 0.3}$ |
| REFINE | $7.9_{\pm 1.1}$ | $42.2_{\pm 1.3}$ | $121.7_{\pm 13.5}$ | $16.4_{\pm 1.5}$ | $12.4_{\pm 2.7}$ | $5.2_{\pm 0.3}$ | $155.5_{\pm 39.9}$ | $0.8_{\pm 0.4}$ |
| GREED | $7.5_{\pm 0.3}$ | $12.3_{\pm 0.9}$ | $15.6_{\pm 0.9}$ | $6.9_{\pm 0.8}$ | $6.8_{\pm 0.3}$ | $8.7_{\pm 0.8}$ | $14.8_{\pm 0.1}$ | $3.8_{\pm 0.5}$ |
| GELATO | $0.2_{\pm 0.2}$ | $98.3_{\pm 0.5}$ | $0.9_{\pm 1.9}$ | $98.9_{\pm 0.2}$ | $0.9_{\pm 0.0}$ | $83.0_{\pm 0.8}$ | $0.7_{\pm 0.1}$ | $77.8_{\pm 0.7}$ |

Table 11: Solution quality of methods with *non-unit* edit costs, w.r.t. nMAE ($\downarrow$) and EHR ($\uparrow$) in %.

**Transfer-learning across datasets** An interesting question is whether the weights learned for one datasets can be used at inference time for another dataset in a transfer learning fashion, or whether the weights are dataset-specific.

An issue to take into account is that for different datasets, node and edge features might have different meanings. For example, the same one-hot encoding which represents a carbon atom in the molhiv-16 dataset represents a type of Python function in the code2-22 dataset. Even across datasets from the same field, the problem persists. Indeed, in the ZINC molecular dataset, atom types are indexed differently form the molhiv-16 dataset. Therefore, a model would be unable to transfer the patterns learned on one dataset to another one.

This issue does not arise in unlabeled datasets, such as LINUX and IMDB-16, as the model exploits only the graph topology. Table 12 reports the EHR ($\uparrow$) in this transfer learning setting on the two unlabeled datasets. We notice that while there is some degradation, likely due to the differences in the graph distributions across the two datasets, the solution quality remains quite satisfactory.

Moreover, to circumvent the issue with features on labeled graphs, we introduce a new molecular dataset, moltox-16. This dataset is obtained from the OGB repository (Hu et al., 2020b), specifically by taking all the graphs with less than 16 nodes from the ogbg-moltox21 dataset. Since molhiv-16 and moltox-16 both are obtained from OGB and belong to the same domain (molecules), they share the same feature space. Table 13 reports the EHR ($\uparrow$) when doing transfer learning between these two molecular datasets. Indeed in this case we can observe that the drop in performance, albeit present, is minimal. This shows that the model is indeed able to apply the molecular patterns learned on one dataset to the other one, in both directions.

Finally, Table 14 reports EHR ($\uparrow$) when doing transfer learning between labeled datasets that do not share a common feature space. As expected, the degradation in solution quality is much more pronounced. In this scenario, in light of the results of Table 5 on limited or noisy supervision, it is better to spend some computational resources for producing a few ground truth matching, and using those to train a specialized model for the data distribution at hand.

| Training | Testing | |
|---|---|---|
| | LINUX | IMDB-16 |
| LINUX | $99.9_{\pm 0.1}$ | $74.8_{\pm 1.4}$ |
| IMDB-16 | $88.5_{\pm 1.1}$ | $99.9_{\pm 0.1}$ |

Table 12: Transfer learning performance w.r.t. EHR ($\uparrow$) on unlabeled graphs from different domains.

| Training | Testing | |
|---|---|---|
| | molhiv-16 | moltox-16 |
| molhiv-16 | $95.3_{\pm 0.8}$ | $97.8_{\pm 0.4}$ |
| moltox-16 | $91.7_{\pm 0.7}$ | $98.1_{\pm 0.3}$ |

Table 13: Transfer learning performance w.r.t. EHR ($\uparrow$) on molecular graphs with shared features.

| Training | Testing | | |
|---|---|---|---|
| | ZINC-16 | molhiv-16 | code2-22 |
| ZINC-16 | $91.1_{\pm 1.1}$ | $19.0_{\pm 0.9}$ | $19.8_{\pm 1.7}$ |
| molhiv-16 | $39.8_{\pm 0.9}$ | $95.3_{\pm 0.8}$ | $58.2_{\pm 1.9}$ |
| code2-22 | $4.6_{\pm 0.7}$ | $4.6_{\pm 0.3}$ | $95.7_{\pm 0.8}$ |

Table 14: Transfer learning performance w.r.t. EHR ($\uparrow$) on labeled graphs with different features.

**Inference times on CPU**  In Table 15, we report the inference running times of GELATO when run on CPU rather than on GPU. Clearly, this results in slower inference times (by up to an order of magnitude). However, inference with GELATO remains faster than other ML-based methods that return matchings, especially for larger graphs. Finally, we remark that runtime considerations are heavily dependent on the system at hand.

**Training times**  We report the training times for all considered learning-based approaches. Table 16 provides the total runtimes of the complete training process for each method using the setups described in Section F. The reported times include both model training and the evaluation conducted during training. Note that variation in runtimes across methods is also influenced by author-proposed default parameter settings, the use of mechanisms such as early stopping, and whether a method supports native parallelization.

| Method | AIDS | LINUX | IMDB-16 | ZINC-16 | molhiv-16 | code2-22 |
|---|---|---|---|---|---|---|
| GPU (H100) | $3.2_{\pm 0.2}$ | $2.9_{\pm 0.1}$ | $3.6_{\pm 0.2}$ | $4.2_{\pm 0.2}$ | $4.1_{\pm 0.1}$ | $5.3_{\pm 0.1}$ |
| CPU | $11.5_{\pm 0.1}$ | $8.2_{\pm 0.2}$ | $24.7_{\pm 0.3}$ | $27.1_{\pm 0.1}$ | $24.9_{\pm 0.2}$ | $61.2_{\pm 0.1}$ |

Table 15: Average inference runtime per graph pair (ms) on GPU and CPU.

Overall, GELATO requires lower training times compared to the other ML-based methods that return solutions, i.e. MATA*, GEDGNN and GEDHOT.

| Method | AIDS | LINUX | IMDB-16 | ZINC-16 | molhiv-16 | code2-22 |
|---|---|---|---|---|---|---|
| GREED | 929 | 610 | 2342 | 2188 | 2885 | 1582 |
| GRAPHEDX | 2806 | 6743 | 523 | 3820 | 7800 | 4962 |
| MATA* | 7592 | 7993 | 11349 | 8247 | 15738 | 75007 |
| GEDGNN | 26852 | 29564 | 13900 | 49694 | 57057 | 83584 |
| GEDHOT | 22148 | 36353 | 14523 | 36506 | 44731 | 56648 |
| GELATO | 5422 | 4760 | 7850 | 9194 | 11979 | 7216 |

Table 16: Total training runtime for the ML-based methods (s).

# I    USE OF LARGE LANGUAGE MODELS

Large Language Models have been used while writing this paper to polish writing (e.g., grammar, spelling, word choice). Moreover, they have been used to write parts of the code (e.g. parsing files, vectorizing tensor operations) used for the experiments.

