# OpenReview forum: "Gelato: Graph Edit Distance via Autoregressive Neural Combinatorial Optimization"
_ICLR.cc/2026/Conference — ICLR 2026 Poster_

### Official Review · Reviewer_jqC1 · 2025-10-28

**Soundness:** 2
**Presentation:** 2
**Contribution:** 2
**Rating:** 4
**Confidence:** 2

**Summary:**

This paper proposes a novel method to approximate the graph edit distance ( GED). The major contribution is the use of a neural network to solve the GED sequentially.

**Strengths:**

The proposed method is interesting as it allows to construct the edit paths.

**Weaknesses:**

The designation of “autoregressive” seems not to be correct, or at best not well justified. The proposed method can be clearly defined as sequential. At the end, it uses roughly a search-based strategy in the same spirit as A* and related methods.

A major issue is that there are no guarantees of optimality for the proposed method. Moreover, it is well known that sequential optimization is non-optimal in general.

While the authors provide some theoretical results, they cannot be explored in practice. For instance, Lemma 1, which motivates the greedy selection, cannot be exploited because the optimal function cannot be computed in practice, making these results intractable.

It would be relevant to provide a comprehensive ablation study beyond Section 5.3. For instance, it is not clear if the batch normalization and the residual connections are beneficial or not. Moreover, the paper does not justify the choice of the values of most hyperparameters, such as the number of layers of the GIN set to 5, embedding dimension, number of randomly selected pairs, resembling with k=32...

The computational complexity needs to be studied in depth. The authors mainly present the inference runtime, but not the training runtime.

The expression “Any state is also a terminal state” is misleading.

There are several spelling and grammatical errors that can be easily identified and corrected, such as “We define a the set…”, as well as GINE.

**Questions:**

No further comments.

---

> ### Author Response · Authors · 2025-11-16
>
> Thank you for your review. Please find detailed answers to your comments below.
>
> >The designation of “autoregressive” seems not to be correct, or at best not well justified.
>
> We agree that "sequential" would also be an appropriate term for the method, and that "autoregressive" is usually used for sequence-based models. However, we argue that also "autoregressive" is correct, as the model iteratively makes a prediction (in this case, a next-match prediction, instead of next-token) based on both the context and its previous outputs (i.e., the previous matches).
>
> Formally, given two graphs $G_1, G_2$, at step $i$ our GNN predicts a node match $\mu_i = (u, v)$, meaning that the two nodes $u \in V_1, v \in V_2$ are matched together. Particularly, we have that $\mu_i = f(G_1, G_2, \\{ \mu_j \ \colon j < i \\})$, i.e. the $i$-th match depends solely on the two graphs and on the nodes that were matched at the previous steps. This differs from one-shot models that predict all matches independently, or models that might change some of the already-chosen matches based on subsequent choices.
>
> However, if the reviewers agree that the term is inaccurate, we would be open to replacing "autoregressive" with "sequential".
>
> > A major issue is that there are no guarantees of optimality for the proposed method.
>
> Please note that GED is NP-hard, and therefore it is extremely unlikely to admit a polynomial-time algorithm with optimality guarantees. Indeed, no machine-learning-based method for GED in the literature yields any optimality guarantees. In fact, to the best of our knowledge, no method in the field of neural combinatorial optimization has optimality guarantees.
>
> As you correctly point out and as we state in the paper, the function of Lemma 1 cannot be efficiently computed. However, with our model we seek an approximation of such a function by learning it from data, e.g., by exploiting reoccurring patterns in the graph distribution at hand. Our approach yields state-of-the-art results that often are close-to-optimal, showcasing that is indeed possible to learn such a good approximation.
>
> Finally, we agree that trading-off computational runtime for optimality guarantees would be a desirable property. We envision that integrating Gelato or similar methods within branch-and-bound schemes could reduce the running times needed to reach optimality. We have added a remark in the conclusions.
>
> > It would be relevant to provide a comprehensive ablation study beyond Section 5.3.
>
> We agree that a more comprehensive hyperparameter study is interesting. Thanks to your suggestion, we have now added a dedicated figure (Fig. 6) and paragraph in the main paper, and a more comprehensive section in the appendix. In particular we observe that:
> - both batch normalization and residual connections have a positive influence on the solution quality, albeit a minor one.
> - The solution quality generally improves with the dimensionality $d$, and shows diminishing returns when moving from $d=128$ to $256$. Clearly, this comes at the expense of an increased number of parameters (e.g. roughly 200k at $d=128$ and 800k at $d=256$). Similar results hold for the number of message passing layers.
> - Ensembling consistently improves solution quality.  Especially on hard datasets, increasing $k$ to, e.g., $128$ yields improvements beyond the results reported in Table 1 for $k=32$. For example, on ZINC-16, it improves the EHR from 91% to 97%. This comes with a slight increase of inference running times (on ZINC-16, from 4.2 milliseconds/pair to 6.3).
>
> > The computational complexity needs to be studied in depth. The authors mainly present the inference runtime, but not the training runtime.
>
> Thank you for the suggestion.
> We have added Table 16 in the appendix, which reports training times, and we reference it in the main text. Note that non-neural baselines have no training time and the "training" of Grail requires calls to the Gemini API, so they were excluded from the table. Overall, Gelato requires lower training times compared to the other ML-based methods that return matchings, i.e. MATA*, GEDGNN and GEDHOT.
>
> > The expression “Any state is also a terminal state” is misleading.
>
> We agree that this expression, due to space constraints, was quite unclear. We clarified the expression as following: *Note that each state $s = (G_1, G_2, \mu)$, even if $\mu$ leaves some nodes unmatched, represents a valid solution to the matching problem between $G_1$ and $G_2$. Indeed, the unmatched nodes belong to the set of source and target deletions $\mu^+$ and $\mu^-$. Therefore, in our search space, any such state is also a valid terminal state.*
>
> > There are several spelling and grammatical errors that can be easily identified and corrected, such as “We define a the set…”, as well as GINE.
>
> Please note that GINE is not a typo, it is the extension of GIN to edge-labeled graphs. We fixed the other error, and some others that we found. Thank you for pointing this out.

---

> > ### Comment · Reviewer_jqC1 · 2025-11-23
> >
> > I thank the authors for their reply, addressing the major raised issues. For this reason and considering the other reviews and feedbacks, I have increased the overall score.

---

> > > ### Author Response · Authors · 2025-11-27
> > >
> > > Dear reviewer. Thank you very much for your quick response and for acknowledging the improvements made in our revised manuscript.

---

### Official Review · Reviewer_eo4E · 2025-10-31

**Soundness:** 4
**Presentation:** 3
**Contribution:** 4
**Rating:** 8
**Confidence:** 3

**Summary:**

GELATO solves the graph edit distance (GED) problem
by reformulating it as a sequential decision making process.
The graph neural network (GNN) is trained to predict one matching node pair at a time,
given a partially matched pair of graphs,
with efficiency and soundness considerations such as
overlapping state space and automorphisms,
allowing for autoregressive inference to generate the full matching
to compute the GED.
Comprehensive experiments show that GELATO achieves high solution quality compared to baselines,
while taking less time, especially compared to approaches with an non-neural matching/search component.
Further analyses include generalization on larger graphs, robustness under limited supervision and ablations.

**Strengths:**

The paper proposes a novel method, with well-motivated and carefully-considered components.

The experiments are comprehensive and the results are strong.

The experiments support the main motivations for this research,
e.g. generalization to larger graphs and faster inference times.

**Weaknesses:**

While the paper is well-written overall,
some more-involved parts of the method could be better explained,
especially the state-space reduction and automorphism considerations.

**Questions:**

1. What is the basis for selection of 3 baselines out of 8 in Section 5.2 / Figure 5?

---

> ### Author Response · Authors · 2025-11-16
>
> Thank you for your review, and for highlighting several strong aspects of our submission. We address your comments below.
>
> > While the paper is well-written overall, some more-involved parts of the method could be better explained, especially the state-space reduction and automorphism considerations.
>
> Thank you for this comment. We added three paragraphs better explaining the state-space reduction. In particular, we added an intuitive explanation at lines 230-234, a clarification on the result of Theorem 1, and expanded the sentence referencing Figure 3 to better illustrate the reduction on two concrete examples.
>
> Moreover, we clarified the automorphism considerations, introducing a sentence at lines 315-317. We now provide a clearer discussion of the visualization in Figure 4 to enhance the understanding of this concept.
>
> > What is the basis for selection of 3 baselines out of 8 in Section 5.2 / Figure 5?
>
> We selected representative methods to avoid cluttering the figure, but we remark that the trend is the same for the remaining methods. In particular, MIP-F2 is consistently better than the other two classical methods (BRANCH and REFINE), and GEDHOT is consistently better than GEDGNN. Finally, MATA*, Greed and GraphEdx were excluded from this experiment. Indeed, the computational requirements for MATA* do not scale to the graph sizes we consider here. The other two methods do not return discrete node matchings and are thus not directly comparable.

---

> > ### Author Response · Authors · 2025-11-27
> >
> > Dear reviewer. Thank you again for your valuable feedback.
> > We hope that our response has adequately addressed all your comments, particularly concerning the clarity. Please let us know if you have any further or follow-up questions so that we can address them in time.

---

### Official Review · Reviewer_F8dj · 2025-11-01

**Soundness:** 2
**Presentation:** 2
**Contribution:** 2
**Rating:** 4
**Confidence:** 5

**Summary:**

This paper addresses the problem of Graph Edit Distance (GED) estimation by framing it as a sequential decision-making task. The authors propose GELATO, an autoregressive framework built on a GNN backbone, in which a graph matching solution is incrementally constructed step-by-step. In line with classical search-based algorithms for GED, the method introduces a dynamic programming–style decomposition of the problem, where partial matchings and their induced subproblems are reduced while preserving equivalence to the optimal solution set. The model is trained using node-level matching supervision and evaluated on several benchmark datasets, with experiments demonstrating improved generalization to larger graph sizes compared to some previous learning-based GED predictors.

**Strengths:**

**Strengths**

- GED prediction often requires resolving symmetries and tie-breaking, further exacerbated by expressivity limitations of standard GNNs. This make the sequential decision framework, in line with  traditional heuristic search methods for GED, well-motivated. I also like the analysis showing that the reduced subproblem maintains equivalence of optimal solutions.

-  Despite relying on explicit node-level matching supervision, which would usually count as a disadvantage, the paper shows robust generalization to larger graphs, which confirms practical utility of the approach.

- By generating incremental matchings, the method offers interpretability advantages compared to one-shot predictors.

**Weaknesses:**

**Weaknesses**

1.  The statement of Theorem 1 differs between the main paper and the appendix, and the notation in the main paper version is unclear. Since the theorem plays a central conceptual role, the paper would benefit from, aligning the statements across sections, clarifying notation, and adding a brief intuitive explanation (2–3 sentences) to the main body to highlight why equivalence is preserved.

2. The architecture section does not clearly specify how partial matchingsare represented internally.  Do cross-graph edges introduced by a partial matching participate in message passing? If so, the graph structure is dynamically modified, which may have unintended representational implications.  If not, how are embeddings of partially matched nodes coupled or tied together?  Similarly, the implementation of the reduce operation in the neural pipeline is not clearly described.

3. The choice of a 128-dimensional embedding for comparatively small graphs is not well justified, since the node embedding matrix is larger than the adjacency matrix!

**Questions:**

Please refer to the weaknesses for some questions.


In addition,  I would like to better understand the practical utility and trade-offs of using an autoregressive approach as opposed to existing one-shot GED predictors. In general, one would expect autoregressive inference to incur higher computational cost, due to stepwise decoding and possible sensitivity to early decisions that may require techniques such as parallel beam search. While the ability to produce interpretable edit paths is a meaningful benefit, the paper does not clearly analyze what is gained and what is sacrificed in moving from one-shot estimation to autoregressive decoding.

In the reported timing comparisons, the authors attribute improved runtime to the fact that GELATO is “GPU-friendly,” whereas several baselines are not. This makes it difficult to disentangle architectural advantages of the autoregressive formulation from implementation differences. It is desirable to have a more direct study of these trade-offs including inference latency, resource usage, prediction accuracy improvements, etc.

Also, is there any reason why GraphEDX (Jain et al., 2024) has been omitted as a baseline. I would  consider it a decent exemplar of GPU-frienly one-shot predictor for comparison.  I suspect  GraphEDX does not scale well to larger graphs. However, a comparison on in-distribution graphs, highlighting accuracy vs. interpretability vs. inference costm, would be intersting.

Minor comment: The authors highlight Jain et al., 2024 in the dataset isomorphism issue, but perhaps [1] would be a better reference in this regard.

[1] Position: Graph Matching Systems Deserve Better Benchmarks. In Forty-second International Conference on Machine Learning Position Paper Track.



Overall, I am positive on the motivation and approach of this paper, but have questions about the neural architecture design and trade-offs around the autoregressive design.

**Details Of Ethics Concerns:**

I do not have any ethics concerns.

---

> ### Author Response · Authors · 2025-11-16
>
> We thank you for the insightful review, and for acknowledging the quality, practical utility and interpretability of our approach. We also thank you for your insights on how the paper could be improved, which we address one-by-one below.
>
> > Statement of Thm 1...
>
> Thanks for pointing it out. We added the missing $\supseteq \mu$, which explicitly indicates that the optimal matching $\mu^*$ is a superset of the fixed matching $\mu$. We agree that additional explanations will make the reduction easier to understand. We added an intuitive explanation at lines 230-234, a clarification of the result of Thm.1, and expanded the sentence referencing Fig 3.
>
> > The architecture section...
>
> Indeed, we represent all matches as additional edges, which take part in the message passing. Thanks to your comment, we now specify this more explicitly in Sect. 4.1, where we describe how to transform an instance to a graph.
> As you correctly point out, this has representational implications, and it is the key component for breaking symmetry and exploiting the information from previous matches.
>
> In principle, the reduce operation could be implemented by actually deleting nodes, but this requires re-indexing the edge_index vector (in the PyG graph). In our implementation, we simply remove all edges adjacent to the deleted nodes, as from the message-passing perspective this is identical. We added a paragraph on the implementation in Sect. E.1, and reference it in the main text.
>
> > 128-dimensional embedding
>
> Note that the embeddings are computed on the graph representations of (partial) matching instances, not directly on the source or target graphs themselves. Thus, even for relatively small graphs, the number of possible instances is typically very large. For a single graph pair, this number is already factorial in the size of the two graphs. Therefore, the dimension of the embeddings is driven by the need to capture the diversity of matching instances across the entire dataset.
>
> However, we agree that the specific choice of $d=128$ requires further justification.
> We have now added a hyperparameter study, including one on the embedding dimensionality (Sect. 5.3 and F). The results show that using $d=32, 64$ yields a significant reduction in solution quality compared to $d=128$. Actually, the solution quality can be slightly improved by increasing $d$ to $256$, albeit at the cost of memory usage (from ~200k parameters to ~800k).
>
> > trade-offs of using autoregressive...
>
> As shown in Table 4, the autoregressive strategy yields a great boost to solution quality. Intuitively, this is because a sequential model can exploit the previous matches to make decisions in the subsequent steps (e.g., two nodes whose neighbors are already matched are more likely to be matched) and break potential symmetries. As you correctly point out, this leads to some overhead due to the stepwise decoding, and due to using ensembling/beams to mitigate the sensitivity to the first choice. However, one-shot prediction models like GEDGNN and GEDHOT have to resort to Hungarian-like algorithms to solve the linear assignment, which are also expensive (and cannot break symmetries). Moreover, these models also use ensembling to find low-cost matchings.
> This ultimately results in Gelato being ~100 times faster, and with significantly higher solution quality.
>
> > timing comparison
>
> One-shot models like GEDGNN or GEDHOT rely on CPU-based Hungarian-like algorithms to obtain assignments. Similarly, MATA* relies on a CPU implementation of A*. Implementing these algorithms to process several small graphs in parallel on GPU seems highly complex, and we are not aware of any such libraries.
>
> On the other hand, Gelato is implemented natively in pytorch and is therefore amenable to GPU computation. We agree that this makes it hard to disentangle where the speed benefits come from. However, we would argue that our efficient implementation is a strength of our method, rather than a weakness. To provide a fairer comparison, we now provide inference times of Gelato also on CPU, reported in Table 15. The results show that inference with Gelato, albeit ~10x slower than on GPU, remains faster than the other ML-based methods that return matchings.
>
> Finally, we remark that while the fast runtime is a positive feature, it is not the main contribution of our work, which is providing a significantly better solution quality compared to existing baselines.
>
> > GraphEDX
>
> We originally omitted GraphEdx because it natively provides only soft matchings (and is thus less interpretable), and we focused primarily on baseline methods that produce discrete node matchings. However, we agree that GraphEdx is a relevant baseline, and we have now included it in the experiments (in the same category as Greed). The results show that, although GraphEdx performs competitively among the baseline methods, it falls short of the performance achieved by Gelato.
>
> > Minor comment
>
> Thanks, we have added this reference.

---

> > ### Author Response · Authors · 2025-11-27
> >
> > Dear reviewer. Thank you again for your valuable feedback.
> > We hope that our response has adequately addressed all your comments and questions. Please let us know if you have any further or follow-up questions so that we can address them in time.

---

### Official Review · Reviewer_W2sn · 2025-11-04

**Soundness:** 3
**Presentation:** 4
**Contribution:** 2
**Rating:** 4
**Confidence:** 3

**Summary:**

In this paper, the authors present Gelato, an RL agent framework for solving graph edit distance. The authors first formulate the editing problem as a matching problem, and then train an RL agent to assign matched node pairs sequentially. The experimental evaluation shows that the proposed RL framework outperforms existing non-RL learning methods.

**Strengths:**

* This paper is well written with a self-contained story.
* The evaluation on various datasets demonstrates the utility of the proposed RL method, and the empirical improvements over existing methods are impressive.

**Weaknesses:**

* This manuscript overlooks an important prior work, [Liu et al: Revocable Deep Reinforcement Learning with Affinity Regularization for Outlier-Robust Graph Matching](https://openreview.net/pdf?id=QjQibO3scV_). Liu et al presented an RL framework for the QAP form of graph matching, which is also used in Gelato. Though the underlying application and datasets are different, the methodologies of both papers seem relevant. I will be more convinced of the contribution and novelty of this paper if the authors can show the technical differences, unique innovations, and insights compared to Liu et al.

**Questions:**

* I will be happy to reconsider this manuscript if the authors could provide more discussion with the relevant work, [Liu et al: Revocable Deep Reinforcement Learning with Affinity Regularization for Outlier-Robust Graph Matching](https://openreview.net/pdf?id=QjQibO3scV_)
* What will the performance look like if the methodology in Liu et al. is directly applied to GED learning?
* If I understand it correctly, different models are trained on different datasets. What will the accuracy look like if a model is trained on, say, the AIDS dataset and tested on other datasets? Is it possible to plot a confusion matrix to report the generalization ability? This question is important because the agent is learning the QAP, and we should expect it to generalize with such a general form.
* Is it possible to train a general agent by mixing multiple training datasets?

---

> ### Author Response · Authors · 2025-11-16
>
> Thank you for your review. Please find point-by-point answers below.
>
> > This manuscript overlooks an important prior work, Liu et al: Revocable Deep Reinforcement Learning with Affinity Regularization for Outlier-Robust Graph Matching. [...]
> I will be happy to reconsider this manuscript if the authors could provide more discussion with the relevant work, Liu et al.
>
> Thank you for this interesting reference. We agree that both approaches share similarities, in particular the step-wise prediction of matches. At the same time, there are several differences with our work. First, note that our work is not strictly-speaking based on reinforcement-learning (RL), as the supervision comes from some ground-truth solutions in the form of a link prediction task, rather than from some reward. Moreover, as discussed in Section B and as you correctly point out, the graph matching/network alignment subfield has a different focus, namely matching few large graphs, often with no ground truths available for training. Furthermore, in Liu et al. and other graph matching papers, the objective function is designed to maximize the cumulative node and edge affinities or similarities, while in the GED literature usually one tries to minimize cumulative edit costs. Moreover, it seems like in Liu et al. there is no reduction of the search space, as allowed by our Thm. 1.   Finally, Liu et al. represents the instances as the product graph (called the association graph) of the two input graphs and casts the match selection as a node-level prediction on this product graph. In contrast, we represent the matches as additional edges between the input graphs, and cast the match selection as a link prediction task.
>
> We now expanded the discussion on additional related work, in Section 1.1, and we include a paragraph on the graph matching field. Here, we acknowledge the similarities and differences with Liu et al. In Section B, we go into more detail. Moreover, we added a reference to this related work in the conclusion, mentioning that the use of RL like in Liu et al. is an interesting research direction for GED learning.
>
> > What will the performance look like if the methodology in Liu et al. is directly applied to GED learning?
>
> The method of Liu et al. seems to be tailored to the graph matching/network alignment field, and thus the objective function is designed to maximize the cumulative node and edge affinities or similarities, rather than minimizing cumulative costs. Due to the complexity of their affinity regularization technique, directly adapting their model to GED predictions would require significant modifications.
> Nonetheless, we believe that parts of the methodology proposed by Liu et al., particularly their use of reinforcement learning, could be very beneficial to the GED field in the future to reduce the need for expensive ground truth computations. We now mention this in the conclusions.
>
> > If I understand it correctly, different models are trained on different datasets. What will the accuracy look like if a model is trained on, say, the AIDS dataset and tested on other datasets? Is it possible to plot a confusion matrix to report the generalization ability?
>
> > Is it possible to train a general agent by mixing multiple training datasets?
>
> Your question is very interesting. In principle, it is possible both to train a general model, as well as to perform transfer learning to other datasets. However, note that because of the NP-hardness of the problem, it is likely impossible to learn a model that would perform well on *any* data distribution.
> Moreover, practically, this model would require that the node and edge features are shared across datasets, which unfortunately does not hold for most of the datasets from the literature. For example, a one-hot encoding might represent a carbon atom in the ZINC dataset, while in the AIDS dataset it might represent a chlorine atom, hindering generalizability across datasets.
>
> That said, we acknowledge the relevance of your point about the generalization abilities of learning-based methods. In response, we now include transfer learning results in Appendix G (Table 12-14). For instance, on unlabelled datasets such as LINUX and IMDB-16 (where only the graph structure matters and there is no issue with labels), results indicate a satisfactory level of generalization.
> Additionally, for molecular datasets that share a consistent feature representation (such as molhiv-16 and the newly introduced moltox-16), we observe very strong generalization performance across datasets.
>
> When feature spaces are not shared across datasets, the transfer learning performance decreases. In this scenario, in light of the result of Table 5, we find it better to spend a few minutes generating some ground truth for the graph distribution at hand rather than use a pre-trained model from another dataset.

---

> > ### Author Response · Authors · 2025-11-27
> >
> > Dear reviewer. Thank you again for your valuable feedback.
> > We hope that our response has adequately addressed all your comments, particularly with regard to the discussion of the work by Liu et al and the transfer learning capabilities of our method. Please let us know if you have any further or follow-up questions so that we can address them in time.

---

### Author Response · Authors · 2025-11-16

Dear reviewers, we would like to express our appreciation for the several positive comments on our paper, including
- The novelty of our approach (eo4E)
- The strong empirical results (W2sn, eo4E)
- The robust generalization to larger graphs (F8dj, eo4E)
- The interpretability of results in form of matchings/edit paths (F8dj, jqC1)

We would like to thank you for your insightful comments and suggestions on how to further improve the paper. We have carefully considered all of your feedback and made revisions accordingly (highlighted in blue in the paper), which we believe have considerably strengthened our paper. In particular, based on your feedback, we included
- clearer explanations (e.g. for the reduction procedure)
- an extensive hyperparameter study
- additional related work

Please find detailed replies to all comments in the individual answers below.

---

### Author Response · Authors · 2025-12-02
**Summary of Discussion and Paper Updates**

Dear AC,
we appreciate your effort in taking on additional work to provide meta-reviews in these unforeseen circumstances.  We provide here a summary of our contributions and of the discussion with reviewers.

**Summary**

In our paper, we provided a model for learning to solve the graph edit distance problem based on a novel autoregressive formulation and backed by theoretically-sound considerations on symmetry and state-space reductions. Empirical results show that our model **largely outperforms** all existing baselines, while being 100x **faster** than ML-based and solution-returning competitors. Moreover, our paper is the **first one** to showcase generalization to larger graph sizes and robustness to noisy or limited supervision.

**Reviewer W2sn**

The reviewer acknowledged that the paper is well-written and that the empirical improvements over existing methods are impressive.
They raised concerns only about one missing related work from the deep graph matching field, which is adjacent to the graph edit distance learning one. Although this method belongs to a different subfield and is thus not directly applicable to our task, we have now extended our already-existing review of the deep graph matching field and discuss the requested paper in our manuscript.
They further asked about the transfer learning performance of our method. We now provide promising results for this in App.G.

The reviewer **did not engage** in discussion before the freeze. However, since we address their only concern, and since they explicitly indicated that in this case they would "be happy to reconsider this manuscript", we are confident they would have increased their score.

**Reviewer F8dj**

The reviewer acknowledged that the sequential decision framework is well-motivated and that our method shows robust generalization to larger graphs, which confirms the practical utility of the approach.
They raised some concerns and questions:
- *Clarity of Thm.1 and of the architecture section.* → We added several paragraphs to make these parts clearer.
- *Chosen embedding dimension might be too large.* → This was addressed by an hyperparameter study that proves that solution quality drops if the dimension $d$ is lower than our default value of 128. In fact, increasing $d$ to 256 yields an **additional boost** to solution quality, at the expense of memory usage.
- *Trade-offs of the autoregressive approach.* → We remark that, as shown in Table 4, the autoregressive strategy yields a boost to solution quality and provided some intuitive explanations. Moreover we remark that our simple design allows to write the code in vectorized pytorch operations, leading to a 100x speedup wrt competitors (on GPU). Since the reviewer requested a fairer comparison with CPU-bound competitors, we have shown that Gelato remains 10x faster even on CPU.
- *Add additional baseline and reference.* → We added the requested baseline (GraphEdX) and the additional reference.

The reviewer also **did not engage** in discussion before the freeze.  Since we address all of their concerns and questions, and since the reviewer mentioned that they are "positive on the motivation and approach of this paper", we are quite confident they would have also increased the score.

**Reviewer eo4E**

The reviewer was **very positive** about our paper, recognizing that the method is novel, the results are strong, that Gelato generalizes to larger graphs, and is faster than competitors.
They asked to better explain the state-space reduction and automorphism considerations, which we now do in the revised manuscript. Moreover, we explained that some baselines were removed from Fig. 5 to avoid cluttering and that the conclusions are not affected by this.

**Reviewer jqC1**

The reviewer recognized the novelty of the method and the value of providing edit paths.
They initially raised several concerns, making them perhaps the most critical reviewer. However, we addressed all of them, which resulted in the reviewer stating that all major issues were resolved and thus **raising their score**, even considering the other reviews and feedbacks.

**Updates to the paper**

- additional paragraphs to explain the state-space, its reduction via Thm.1, the model architecture, and the automorphism considerations
- additional baseline (GraphEdX) and related work
- a hyperparameter study on embedding dimension, number of layers, and ensemble size, and an ablation study on residual connections and batch normalizations
- statistics on inference times on CPU, and on training times
- experiments on transfer learning performance across datasets

In conclusion, we think that all the concerns raised by reviewers were minor or fully addressable. We believe that our manuscript has improved significantly by addressing all of these comments. We are therefore confident that also the two missing reviewers would have acknowledged the improvement and increased their scores, if given the time to do so.

---

### Meta-Review · Area_Chair_eqpf · 2025-12-20

**Summary:**

The paper introduces GELATO, a Graph Neural Network (GNN) framework for approximating Graph Edit Distance (GED) through a sequential (autoregressive) decision-making process. Unlike traditional one-shot neural predictors that learn static assignment costs, GELATO constructs solutions step-by-step, conditioning each node-match prediction on the history of previous matches. The authors leverage a theoretically-sound state-space reduction (Theorem 1) to shrink subproblems and handle automorphisms to avoid contradictory training signals.

The proposed method demonstrates good empirical gains, often achieving results an order of magnitude better than the second-best baseline in terms of normalized Mean Average Error (nMAE). Its ability to generalize to graphs larger than those seen during training—a chronic pain point in GED research—is particularly noteworthy. While reviewers initially questioned the computational overhead of an autoregressive approach and the lack of optimality guarantees, the authors' rebuttal clarified that the GPU-friendly implementation is actually two orders of magnitude faster than competing ML methods.

**Reviewer Concerns:**

Addressed concerns:

-  Architecture & Thm 1 Clarity: Authors added intuitive explanations for state-space reduction and clarified that partial matches are represented as cross-graph edges participating in message passing.
- Hyperparameters & Training Time: Rebuttal provided Table 16 for training times and a new hyperparameter study (Figure 6) justifying the embedding dimension and layer choices.
- Related Work & Transfer Learning: Authors integrated a discussion of related deep graph matching work (Liu et al.) and provided new transfer learning results (Tables 12-14) in the appendix.

Outstanding concerns:

- Optimality Guarantees: The reviewer remained concerned about the lack of guarantees. While the authors correctly argued that GED is NP-hard , the point that sequential optimization is non-optimal in a general sense remains a formal limitation of the approach.

**Reviewer Scores:**

- W2sn (4 to 6): The reviewer's primary concern was missing related work and transfer learning. Since the authors addressed both with new experiments and discussion, and the reviewer stated they were "happy to reconsider", a score increase is highly likely.
- F8dj (4 to 6): This reviewer was "positive on the motivation and approach" but had technical questions. The authors clarified the neural architecture and provided a CPU-based runtime comparison, addressing the core of the critique.
- eo4E (8): Remained highly positive throughout.
- jqC1 (4 to 6): This reviewer explicitly stated that all "major issues were resolved" and increased their score.

---

### Decision · Program_Chairs · 2026-01-26

Accept (Poster)